# TEST-TIME ADAPTATION FOR REGRESSION BY SUBSPACE ALIGNMENT

**Kazuki Adachi**[*‡] **Shin'ya Yamaguchi**[*†] **Atsutoshi Kumagai**[*] **Tomoki Hamagami**[‡]
[*]NTT Corporation  [†]Kyoto University  [‡]Yokohama National University
{kazuki.adachi,shinya.yamaguchi,atsutoshi.kumagai}@ntt.com
hamagami@ynu.ac.jp

## ABSTRACT

This paper investigates *test-time adaptation (TTA) for regression*, where a regression model pre-trained in a source domain is adapted to an unknown target distribution with unlabeled target data. Although regression is one of the fundamental tasks in machine learning, most of the existing TTA methods have classification-specific designs, which assume that models output class-categorical predictions, whereas regression models typically output only single scalar values. To enable TTA for regression, we adopt a feature alignment approach, which aligns the feature distributions between the source and target domains to mitigate the domain gap. However, we found that naive feature alignment employed in existing TTA methods for classification is ineffective or even worse for regression because the features are distributed in a small subspace and many of the raw feature dimensions have little significance to the output. For an effective feature alignment in TTA for regression, we propose *Significant-subspace Alignment (SSA)*. SSA consists of two components: subspace detection and dimension weighting. Subspace detection finds the feature subspace that is representative and significant to the output. Then, the feature alignment is performed in the subspace during TTA. Meanwhile, dimension weighting raises the importance of the dimensions of the feature subspace that have greater significance to the output. We experimentally show that SSA outperforms various baselines on real-world datasets. The code is available at https://github.com/kzkadc/regression-tta.

## 1 INTRODUCTION

Deep neural networks have achieved remarkable success in various tasks (LeCun et al., 1998a; Krizhevsky et al., 2012; He et al., 2016; Dargan et al., 2020). In particular, regression, which is one of the fundamental tasks in machine learning, is widely used in practical tasks such as human pose estimation or age prediction (Lathuilière et al., 2019). The successes of deep learning have usually relied on the assumption that the training and test datasets are sampled from an i.i.d. distribution. In the real world, however, such an assumption is often invalid since the test data are sampled from distributions different from the training one due to distribution shifts caused by changes in environments. The performance of these models thus deteriorates when a distribution shift occurs (Hendrycks & Dietterich, 2019; Recht et al., 2019). To address this problem, *test-time adaptation (TTA)* (Liang et al., 2023) has been studied. TTA aims at adapting a model pre-trained in a source domain (training environment) to the target domain (test environment) with only unlabeled target data. However, most of the existing TTA methods are designed for classification; that is, TTA for regression has not been explored much (Liang et al., 2023). Regarding TTA for classification, two main approaches have been explored: entropy minimization and feature alignment.

The entropy minimization approach was introduced by Wang et al. (2021), and the subsequent methods follow this approach (Zhou & Levine, 2021; Niu et al., 2022; Zhang et al., 2022; Zhao et al., 2023; Enomoto et al., 2024). Although entropy is a promising proxy of the performance on the target domain, entropy minimization is classification-specific because it assumes that a model directly outputs predictive distributions, *i.e.*, a probability for each class. In contrast, typical regression models output only single scalar values, not distributions. Thus, we cannot use the entropy minimization approach for regression models.

Table 1: Number of valid (having non-zero variance) feature dimensions and feature subspace dimensions (*i.e.*, the rank of the feature covariance matrix), and $R^2$ scores on the test datasets. Although the original feature space has 2048 dimensions in the experiment except for the California Housing dataset, the features of the regression models are distributed within the subspaces that have less than a hundred dimensions. Our method improves $R^2$ scores by the feature subspace in most cases, whereas the naive feature alignment sometimes diverged (displayed as '-') because of too few valid dimensions. See Section 4.3.1 for more details.

| Dataset | #Valid dims. | #Subspace dims. | $R^2$ (↑) | | |
|---|---|---|---|---|---|
| | | | Source | Naive feature alignment | SSA (ours) |
| SVHN | 353 | 14 | 0.406 | - | **0.511** |
| UTKFace | 2041 | 76 | 0.020 | 0.705 | **0.731** |
| Biwi Kinect* | 713 | 34.5 | 0.706 | 0.753 | **0.778** |
| California Housing (100 dims.) | 45 | 40 | 0.605 | - | **0.639** |

*The average over a model trained on each gender/target is reported.

Another approach, the feature alignment, preliminarily computes the statistics of intermediate features of the source dataset after pre-training in the source domain (Ishii & Sugiyama, 2021; Kojima et al., 2022; Eastwood et al., 2022; Adachi et al., 2023; Jung et al., 2023). Then, upon moving to the target domain, the feature distribution of the target data is aligned with the source distribution by matching the target feature statistics with the pre-computed source ones without accessing the source dataset. Although this approach seems applicable to regression because it allows arbitrary forms of the model output, it assumes to use all feature dimensions to be aligned, and does not sufficiently consider the nature of regression tasks. For instance, regression models trained with standard mean squared error (MSE) loss tend to make features less diverse than classification models do (Zhang et al., 2023). In particular, we experimentally observed that the features of a trained regression model are distributed in only a small subspace of the entire feature space (Table 1). In this sense, naively aligning all feature dimensions makes the performance suboptimal or even be harmful in regression as shown in Table 1 because it equally treats important feature dimensions and degenerated unused ones.

In this paper, we address TTA for regression on the basis of the feature alignment approach. To resolve the aforementioned feature alignment problem in TTA for regression, we propose *Significant-subspace Alignment (SSA)*. SSA consists of two components: *subspace detection* and *dimension weighting*. Subspace detection uses principal component analysis (PCA) to find a subspace of the feature space in which the features are concentrated. This subspace is representative and significant to the model output. Then, we perform feature alignment within this subspace, which improves the effectiveness and stability of TTA by focusing only on valid feature dimensions in the subspace. Further, in regression, a feature vector is finally projected onto a one-dimensional line so as to output a scalar value. Thus, the subspace dimensions that have an effect on the line need a precise feature alignment. To do so, dimension weighting raises the importance of the subspace dimensions with respect to their effect on the output.

We conducted experiments on various regression tasks, such as UTKFace (Zhang et al., 2017), Biwi Kinect (Fanelli et al., 2013), and California Housing (Nugent, 2017). The results showed that our SSA retains the important feature subspace during TTA and outperforms existing TTA baselines that were originally designed for classification by aligning the feature subspace.

## 2 PROBLEM SETTING

We consider a setting with a neural network regression model $f_\theta : \mathcal{X} \to \mathbb{R}$ pre-trained on a labeled source dataset $\mathcal{S} = \{(\mathbf{x}_i^s, y_i^s) \in \mathcal{X} \times \mathbb{R}\}_{i=1}^{N_s}$, where $\mathbf{x}_i^s$ and $y_i^s$ are an input and its label, and $\mathcal{X}$ is the input space. Our goal is to adapt $f_\theta$ to the target domain by using an unlabeled target dataset $\mathcal{T} = \{\mathbf{x}_i^t \in \mathcal{X}\}_{i=1}^{N_t}$ without accessing $\mathcal{S}$. Note that the target labels $y_i^t \in \mathbb{R}$ are not available. In the source dataset $\mathcal{S}$, the data $\{(\mathbf{x}_i^s, y_i^s)\}$ are sampled from the source distribution $p_s$ over $\mathcal{X} \times \mathbb{R}$. In the target dataset $\mathcal{T}$, we assume covariate shift (Shimodaira, 2000), which is a distribution shift that often occurs in the real world. In other words, the target data $\mathbf{x}_i^t$ are sampled from the target distribution $p_t$ over $\mathcal{X}$ that is different from $p_s$, but the predictive distribution is the same, *i.e.*, $p_s(\mathbf{x}) \neq p_t(\mathbf{x})$ and $p_s(y|\mathbf{x}) = p_t(y|\mathbf{x})$.

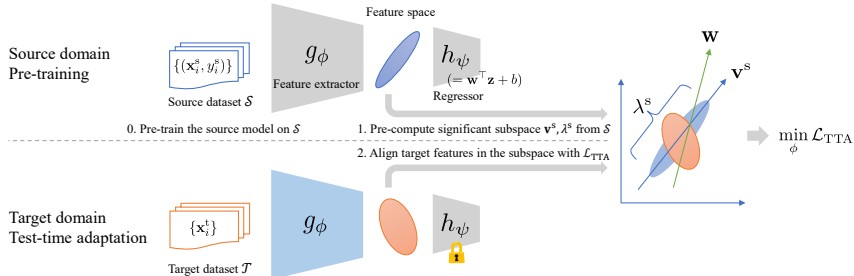

Figure 1: Overview of significant-subspace alignment (SSA). A more detailed procedure is listed in Algorithm 1.

We split the regression model $f_\theta$ into a feature extractor $g_\phi : \mathcal{X} \to \mathbb{R}^D$ and linear regressor $h_\psi(\mathbf{z}) = \mathbf{w}^\top \mathbf{z} + b$, where $D$ is the number of feature dimensions, $\mathbf{w} \in \mathbb{R}^D$, $b \in \mathbb{R}$, $\phi$ and $\psi = (\mathbf{w}, b)$ are the parameters of the models. The whole regression model using the feature extractor and linear regressor is denoted by $f_\theta = h_\psi \circ g_\phi$, where $\theta = (\phi, \psi)$.

## 3 TEST-TIME ADAPTATION FOR REGRESSION

In this section, we describe the basic idea behind *Significant-subspace Alignment (SSA)* in Section 3.1 and describe it in detail in Section 3.2. Our method can be applied regardless of the form of input data since SSA does not rely on input-specific method, such as image data augmentations or self-supervised tasks.

### 3.1 BASIC IDEA: FEATURE ALIGNMENT

The basic idea of our TTA method for regression is to align the feature distributions of the source and target domains instead of using entropy minimization, as usually done in TTA for classification. As we assume a covariate shift where the input distribution changes, we update the feature extractor $g_\phi$ to pull back the target feature distribution to the source one. Here, we describe a naive implementation of the idea and its problem.

First, in the source domain, we compute the source feature statistics (mean and variance of each dimension) on $\mathcal{S}$ after the source training:

$$\boldsymbol{\mu}^{\mathrm{s}} = \frac{1}{N_{\mathrm{s}}} \sum_{i=1}^{N_{\mathrm{s}}} \mathbf{z}_i^{\mathrm{s}}, \quad \boldsymbol{\sigma}^{\mathrm{s}\,2} = \frac{1}{N_{\mathrm{s}}} \sum_{i=1}^{N_{\mathrm{s}}} (\mathbf{z}_i^{\mathrm{s}} - \boldsymbol{\mu}^{\mathrm{s}}) \odot (\mathbf{z}_i^{\mathrm{s}} - \boldsymbol{\mu}^{\mathrm{s}}), \quad (1)$$

where $\mathbf{z}_i^{\mathrm{s}} = g_\phi(\mathbf{x}_i^{\mathrm{s}}) \in \mathbb{R}^D$ is a source feature and $\odot$ is the element-wise product.

Then, we move to the target domain, where we cannot access the source dataset $\mathcal{S}$. Given a target mini-batch $\mathcal{B} = \{\mathbf{x}_i^{\mathrm{t}}\}_{i=1}^B$ sampled from $\mathcal{T}$, we compute the mini-batch mean and variance $\hat{\boldsymbol{\mu}}^{\mathrm{t}}$ and $\hat{\boldsymbol{\sigma}}^{\mathrm{t}\,2}$ analogously to Equation (1).

For feature alignment, we seek to make the target statistics similar to the source ones. For this purpose, we use the KL divergence as Nguyen et al. (2022) proved that it is included in an upper bound of the target error in unsupervised domain adaptation. Concretely, we minimize the KL divergence between two diagonal Gaussian distributions $\mathcal{N}(\boldsymbol{\mu}^{\mathrm{s}}, \boldsymbol{\sigma}^{\mathrm{s}\,2})$ and $\mathcal{N}(\hat{\boldsymbol{\mu}}^{\mathrm{t}}, \hat{\boldsymbol{\sigma}}^{\mathrm{t}\,2})$:

$$\mathcal{L}_{\mathrm{TTA}}(\phi) = \sum_{d=1}^{D} D_{\mathrm{KL}}\left(\mathcal{N}(\mu_d^{\mathrm{s}}, \sigma_d^{\mathrm{s}\,2}) \| \mathcal{N}(\hat{\mu}_d^{\mathrm{t}}, \hat{\sigma}_d^{\mathrm{t}\,2})\right) + D_{\mathrm{KL}}\left(\mathcal{N}(\hat{\mu}_d^{\mathrm{t}}, \hat{\sigma}_d^{\mathrm{t}\,2}) \| \mathcal{N}(\mu_d^{\mathrm{s}}, \sigma_d^{\mathrm{s}\,2})\right), \quad (2)$$

where the subscripts $d$ represent the $d$-th elements of the mean and variance vectors. Here, we used both directions of the KL divergence because it empirically had good results as recommended by Nguyen et al. (2022). The KL divergence between two univariate Gaussians can be written in a closed form (Duchi, 2007)

$$D_{\mathrm{KL}}\left(\mathcal{N}(\mu_1, \sigma_1^2) \| \mathcal{N}(\mu_2, \sigma_2^2)\right) = \left[\log\left(\sigma_2^2/\sigma_1^2\right) + \left\{(\mu_1 - \mu_2)^2 + \sigma_1^2\right\}/\sigma_2^2 - 1\right]/2. \quad (3)$$

However, in regression models, the features tend to be less diverse than in classification (Zhang et al., 2023). In addition to the insight from Zhang et al. (2023), we observed that the features of a regression model trained on $\mathcal{S}$ are distributed in only a small subspace of the feature space and many dimensions of the feature space had zero variances. Table 1 shows the numbers of valid (having non-zero variance) feature dimensions and feature subspace dimensions (see Section 4.3.1 for more details). This property makes the naive feature alignment described above unstable since the KL divergence in Equation (3) includes the variance in the denominator. Also, this naive feature alignment is ineffective because many of the feature dimensions have a small effect on the subspace.

## 3.2 SIGNIFICANT-SUBSPACE ALIGNMENT

In this section, we describe our method, Significant-subspace Alignment (SSA), to tackle the aforementioned problem of naive feature alignment. Figure 1 shows an overview of SSA. As described in Section 3.1, the features of a regression model tend to be distributed in a small subspace of the feature space. Thus, we introduce *subspace detection* to detect a subspace that is representative and significant to the output and then perform feature alignment in the subspace. Subspace detection is similar to principal component analysis (PCA). Further, in our regression model, a feature $\mathbf{z} = g_\phi(\mathbf{x})$ is projected onto a one-dimensional line determined by $\mathbf{w}$ of $h_\psi$ in order to output a scalar value $\mathbf{w}^\top \mathbf{z} + b$. Thus, a subspace basis $\mathbf{v}_d^{\mathrm{s}}$ whose direction is not orthogonal to $\mathbf{w}$ needs precise feature alignment. We use *dimension weighting* to prioritize such dimensions.

**Subspace detection.** After the training on the source dataset $\mathcal{S}$, we detect the subspace in which the source features are distributed. Instead of computing the variance of each dimension as Equation (1), we compute the covariance matrix:

$$\mathbf{\Sigma}^{\mathrm{s}} = \frac{1}{N_{\mathrm{s}}} \sum_{i=1}^{N_{\mathrm{s}}} (\mathbf{z}_i^{\mathrm{s}} - \boldsymbol{\mu}^{\mathrm{s}})(\mathbf{z}_i^{\mathrm{s}} - \boldsymbol{\mu}^{\mathrm{s}})^\top, \qquad (4)$$

where the source mean vector $\boldsymbol{\mu}^{\mathrm{s}}$ is the same as Equation (1).

Then, we detect the source subspace. On the basis of PCA, the subspace is spanned by the eigenvectors of the covariance matrix $\mathbf{\Sigma}^{\mathrm{s}}$, denoted by $\mathbf{v}_k^{\mathrm{s}}$ ($\|\mathbf{v}_k^{\mathrm{s}}\|_2 = 1$). The corresponding eigenvalues $\lambda_k^{\mathrm{s}}$ represent the variance of the source features along the direction $\mathbf{v}_k^{\mathrm{s}}$. We use the top-$K$ largest eigenvalues $\lambda_1^{\mathrm{s}}, \ldots, \lambda_K^{\mathrm{s}}$ ($\lambda_1^{\mathrm{s}} > \cdots > \lambda_K^{\mathrm{s}}$), the corresponding source bases $\mathbf{v}_1^{\mathrm{s}}, \ldots, \mathbf{v}_K^{\mathrm{s}}$, and the source mean $\boldsymbol{\mu}^{\mathrm{s}}$ as the source statistics.

**Dimension weighting.** Since we assume that the output is computed with a linear regressor $h_\psi(\mathbf{z}) = \mathbf{w}^\top \mathbf{z} + b$, the effect of a subspace dimension $\mathbf{v}_d^{\mathrm{s}}$ to the output is determined by $\mathbf{w}^\top \mathbf{v}_d^{\mathrm{s}}$. To prioritize the subspace dimensions that have larger effect on the output, we determine the weight of each subspace dimension as follows:

$$\alpha_d = 1 + |\mathbf{w}^\top \mathbf{v}_d^{\mathrm{s}}|. \qquad (5)$$

$\alpha_d$ assigns a larger weight when the direction along a subspace basis $\mathbf{v}_d^{\mathrm{s}}$ affects the output, or keep the weight to one when not.

**Feature alignment.** This step is done in the target domain. Given a target mini-batch $\mathcal{B}$ sampled from the target dataset $\mathcal{T}$, we project the target features $\mathbf{z}_i^{\mathrm{t}} = g_\phi(\mathbf{x}_i^{\mathrm{t}})$ into the source subspace and then compute the feature alignment loss. The projection of the target feature is computed as follows:

$$\tilde{\mathbf{z}}_i^{\mathrm{t}} = \mathbf{V}^{\mathrm{s}}(\mathbf{z}_i^{\mathrm{t}} - \boldsymbol{\mu}^{\mathrm{s}}), \qquad (6)$$

where $\mathbf{V}^{\mathrm{s}} = [\mathbf{v}_1^{\mathrm{s}}, \ldots, \mathbf{v}_K^{\mathrm{s}}]^\top \in \mathbb{R}^{K \times D}$. With $\tilde{\mathbf{z}}_i^{\mathrm{t}} \in \mathbb{R}^K$, we compute the projected target mean and variance over the mini-batch analogously to Equation (1); this is denoted by $\tilde{\boldsymbol{\mu}}^{\mathrm{t}}$ and $\tilde{\boldsymbol{\sigma}}^{\mathrm{t}\,2}$. On the other hand, the projected source mean and variance are $\mathbf{0}$ and the eigenvalues $\boldsymbol{\lambda}^{\mathrm{s}} = [\lambda_1^{\mathrm{s}}, \ldots, \lambda_K^{\mathrm{s}}]$ since $\mathbf{\Sigma}^{\mathrm{s}} \mathbf{v}_k^{\mathrm{s}} = \lambda_k^{\mathrm{s}} \mathbf{v}_k^{\mathrm{s}}$. Thus, the KL divergence in the detected subspace is computed between two $K$-dimensional diagonal Gaussians $\mathcal{N}(\mathbf{0}, \boldsymbol{\lambda}^{\mathrm{s}})$ and $\mathcal{N}(\tilde{\boldsymbol{\mu}}^{\mathrm{t}}, \tilde{\boldsymbol{\sigma}}^{\mathrm{t}\,2})$. Using subspace detection and dimension weighting, the loss of SSA is:

$$\begin{aligned}
\mathcal{L}_{\mathrm{TTA}}(\phi) &= \sum_{d=1}^{K} \alpha_d \left\{ D_{\mathrm{KL}}\left(\mathcal{N}(0, \lambda_d^{\mathrm{s}}) \| \mathcal{N}(\tilde{\mu}_d^{\mathrm{t}}, \tilde{\sigma}_d^{\mathrm{t}\,2})\right) + D_{\mathrm{KL}}\left(\mathcal{N}(\tilde{\mu}_d^{\mathrm{t}}, \tilde{\sigma}_d^{\mathrm{t}\,2}) \| \mathcal{N}(0, \lambda_d^{\mathrm{s}})\right) \right\} \\
&= \frac{1}{2} \sum_{d=1}^{K} \alpha_d \left( \frac{(\tilde{\mu}_d^{\mathrm{t}})^2 + \lambda_d^{\mathrm{s}}}{\tilde{\sigma}_d^{\mathrm{t}\,2}} + \frac{(\tilde{\mu}_d^{\mathrm{t}})^2 + \tilde{\sigma}_d^{\mathrm{t}\,2}}{\lambda_d^{\mathrm{s}}} - 2 \right).
\end{aligned} \qquad (7)$$

During TTA, we optimize the feature extractor $g_\phi$ to minimize $\mathcal{L}_{\text{TTA}}$, *i.e.*, we seek $\phi^* = \min_\phi \mathcal{L}_{\text{TTA}}(\phi)$. We update only the affine parameters $\gamma$ and $\beta$ of the normalization layers such as batch normalization (Ioffe & Szegedy, 2015) or layer normalization (Ba et al., 2016) inspired by Tent (Wang et al., 2021). This strategy is effective not only to retain the source knowledge but also to enable flexible adaptation (Frankle et al., 2021; Burkholz, 2024). The procedure of SSA is listed in Algorithm 1 of the Appendix.

**Is diagonal Gaussian distribution appropriate?** For computing KL divergence of $\mathcal{L}_{\text{TTA}}$, we assume the source and target feature distributions as diagonal Gaussian. This is reasonable because features are likely to follow a Gaussian distribution when projected onto the feature subspace detected by subspace detection as the number of original feature dimensions increases by the central limit theorem, as described in Figure 3 and Section 4.3.4. Moreover, since subspace detection uses the PCA, the features projected onto the subspace are decorrelated. Thus, assuming that each dimension is independent, *i.e.*, diagonal, is also reasonable.

## 4 EXPERIMENT

This section provides empirical analysis of feature subspaces and evaluations of SSA on various regression tasks. First, we checked whether the learned features are distributed in a small subspace (Section 4.3.1) and then evaluated the regression performance (Sections 4.3.2 and 4.3.3). We also analyzed the effect of the TTA methods from the perspective of the feature subspace (Section 4.3.4).

### 4.1 DATASET

We used regression datasets with two types of covariate shift, *i.e.*, domain shift and image corruption. **SVHN-MNIST**. SVHN (Netzer et al., 2011) and MNIST (LeCun et al., 1998b) are famous digit-recognition datasets. Although they are mainly used for classification, we used them for regression by training models to directly output a scalar value of the label. We used SVHN and MNIST as the source and target domains, respectively.
**UTKFace (Zhang et al., 2017)**. UTKFace is a dataset consisting of face images. The task is to predict the age of the person in an input image. For the source model, we trained models on the original UTKFace images. For the target domain, we added corruptions such as noise or blur to the images. The types of corruption were the same as those of ImageNet-C (Hendrycks & Dietterich, 2019). We applied 13 types of corruption at the highest severity level of the five levels.
**Biwi Kinect (Fanelli et al., 2013)**. Biwi Kinect is a dataset consisting of person images. The task is to predict the head pose of the person in an input image in terms of pitch, yaw, and roll angles. We separately trained models to predict each angle. The source and target domains are the gender of the person in the image. We conducted experiments on six combinations of the source/target gender and task, *i.e.*, {male $\rightarrow$ female, female $\rightarrow$ male} $\times$ {pitch, yaw, roll}. We trained regression models to directly output head pose angles in radian, which are roughly in $(-0.4\pi, 0.4\pi)$.
**California Housing (Nugent, 2017)**. California Housing is a tabular dataset aiming at predicting housing prices from the information of areas. We split the dataset into non-coastal and coastal areas for the source and target domains in accordance with He et al. (2024).

More details of the datasets are provided in Appendix C.1.

### 4.2 SETTING

**Source model.** We used ResNet-26 (He et al., 2016) for SVHN, ResNet-50 for UTKFace and Biwi Kinect, and an MLP for California Housing. We modified the last fully-connected layer to output single scalar values and trained the models with the standard MSE loss on each dataset and task. The details of the training are provided in Appendix C.
**Test-time adaptation with SSA (ours)**. We minimized $\mathcal{L}_{\text{TTA}}$ on the target datasets. We used the outputs of the penultimate layer of the model as features, which had 2048 dimensions. We set the number of dimensions of the feature subspace to $K = 100$ as the default throughout the experiments. More detailed settings are provided in Appendix C.3.
**Baseline.** Since there are no TTA baselines designed for regression, we compared SSA with TTA methods designed for classification but can be naively modified to regression: *Source* (no adaptation), *BN-adapt* (Benz et al., 2021), *Feature restoration (FR)* (Eastwood et al., 2022), *Prototype*,

Table 2: Test scores on SVHN-MNIST. The best scores are **bolded**.

| Method | $R^2 (\uparrow)$ | RMSE $(\downarrow)$ | MAE $(\downarrow)$ |
|---|---|---|---|
| Source | 0.406 | 2.232 | 1.608 |
| DANN | $0.307_{\pm 0.09}$ | $2.406_{\pm 0.16}$ | $1.489_{\pm 0.09}$ |
| TTT | $0.288_{\pm 0.02}$ | $2.443_{\pm 0.03}$ | $1.597_{\pm 0.03}$ |
| BN-adapt | $0.396_{\pm 0.00}$ | $2.251_{\pm 0.01}$ | $1.458_{\pm 0.00}$ |
| Prototype | $0.491_{\pm 0.00}$ | $2.065_{\pm 0.01}$ | $1.479_{\pm 0.01}$ |
| FR | $0.369_{\pm 0.01}$ | $2.300_{\pm 0.02}$ | $1.631_{\pm 0.02}$ |
| VM | $-685.1_{\pm 27.63}$ | $75.83_{\pm 1.52}$ | $75.78_{\pm 1.52}$ |
| RSD | $0.252_{\pm 0.12}$ | $2.497_{\pm 0.20}$ | $1.703_{\pm 0.20}$ |
| SSA (ours) | $\mathbf{0.511}_{\pm 0.03}$ | $\mathbf{2.024}_{\pm 0.06}$ | $\mathbf{1.209}_{\pm 0.04}$ |
| Oracle | $0.874_{\pm 0.00}$ | $1.028_{\pm 0.00}$ | $0.575_{\pm 0.00}$ |

Table 3: Test $R^2$ score and RMSE on California Housing.

| Method | $R^2 (\uparrow)$ | RMSE $(\downarrow)$ | MAE $(\downarrow)$ |
|---|---|---|---|
| Source | 0.605 | 0.684 | 0.516 |
| BN-adapt | $0.318_{\pm 0.00}$ | $0.899_{\pm 0.00}$ | $0.699_{\pm 0.00}$ |
| Prototype | $-0.726_{\pm 0.01}$ | $1.431_{\pm 0.00}$ | $1.196_{\pm 0.00}$ |
| FR | $0.510_{\pm 0.01}$ | $0.762_{\pm 0.01}$ | $0.534_{\pm 0.01}$ |
| RSD | - | - | - |
| SSA (ours) | $\mathbf{0.639}_{\pm 0.00}$ | $\mathbf{0.655}_{\pm 0.00}$ | $\mathbf{0.469}_{\pm 0.00}$ |
| Oracle | $0.729_{\pm 0.00}$ | $0.567_{\pm 0.00}$ | $0.404_{\pm 0.00}$ |

*Variance minimization (VM)*, and *RSD* (Chen et al., 2021). In addition, we used the following methods other than TTA as baselines: *test-time training (TTT)* (Sun et al., 2020), *DANN* (Ganin et al., 2016), and *Oracle* (fine-tuning using labels; performance upper bound). The details of the baseline methods are described in Appendix C.3.

## 4.3 RESULT

### 4.3.1 NUMBER OF DIMENSIONS OF THE FEATURE SUBSPACE

After the pre-training on the source dataset, we counted the numbers of valid feature dimensions (*i.e.*, having non-zero variances) and dimensions of the feature subspace in which the source features are distributed. The latter value corresponds to the rank of the covariance matrix of the source features in Equation (4). Table 1 shows the result of each source dataset and regression test $R^2$ scores. Although the number of feature dimensions is 2048 in ResNet, many feature dimensions of the regression models have zero variance because of ReLU activation. This is the cause of the failure of the naive feature alignment, as described in Section 3.1. Moreover, the source features are distributed in only a small subspace with fewer than a hundred dimensions. In the California Housing dataset, we can also see the same tendency that the number of the subspace dimensions is only 40 whereas the number of the entire feature dimensions is 100. This property limits the performance of the naive feature alignment in regression since aligning the entire feature space is ineffective to the subspace in which the features are actually distributed.

In the MLP used for the California Housing dataset, the subspace dimensions compared to the entire feature space is higher than the ResNets used for the other datasets. One explanation for this difference is model capacity and task complexity. MLP's capacity is low relative to the task (California Housing)'s complexity. On the other hand, the ResNet-26, which has high capacity, resulted in lower subspace dimensions on SVHN, which has low complexity.

### 4.3.2 REGRESSION PERFORMANCE

We evaluated the regression performance in terms of the $R^2$ score (coefficient of determination), which is widely used in regression tasks (see Appendix B). Tables 2 and 3 show the scores for the SVHN-MNIST and California Housing. In the both cases, SSA outperformed the baselines; some of them even underperformed the Source. This is because the baselines were designed for classification tasks and they broke the feature subspaces learned by the source model (see Section 4.3.4). On the other hand, RSD (Chen et al., 2021) is originally designed for regression in UDA but did not work in the California Housing dataset because of numerical instability of SVD performed on every target feature batch. In contrast, our method is stable since it avoids degenerated dimensions during TTA. Table 4 shows the $R^2$ scores on the UTKFace with image corruption. We can see that SSA had the highest $R^2$ scores for most of the corruption types. In particular, SSA outperformed the baselines by a large margin on noise-type corruption which significantly degraded the performance of Source. Table 5 shows the $R^2$ scores on Biwi Kinect with genders different from the source domains. SSA constantly had higher $R^2$ scores than the baselines; the baselines' scores sometimes significantly dropped or even diverged (Prototype). In summary, SSA consistently improved the scores whereas

Table 4: Test $R^2$ scores on UTKFace with image corruption (higher is better). The best scores are **bolded**.

| Method | Defocus blur | Motion blur | Zoom blur | Contrast | Elastic transform | Jpeg comp. | Pixelate | Gaussian noise | Impulse noise | Shot noise | Brightness | Fog | Snow | Mean |
|---|---|---|---|---|---|---|---|---|---|---|---|---|---|---|
| Source | 0.410 | 0.159 | 0.658 | −3.906 | 0.711 | 0.069 | 0.595 | −2.536 | −2.539 | −2.522 | 0.661 | −0.029 | −0.544 | −0.678 |
| DANN | 0.512 | 0.586 | 0.637 | −0.720 | 0.729 | 0.698 | 0.807 | −4.341 | −3.114 | −3.744 | 0.590 | −0.131 | −0.425 | −0.609 |
| TTT | 0.748 | 0.761 | 0.773 | 0.778 | 0.826 | 0.772 | 0.861 | 0.525 | 0.532 | 0.477 | 0.775 | 0.397 | 0.493 | 0.671 |
| BN-Adapt | 0.727 | 0.759 | 0.763 | 0.702 | 0.826 | 0.778 | 0.850 | 0.510 | 0.510 | 0.446 | 0.790 | 0.392 | 0.452 | 0.654 |
| Prototype | −1.003 | −1.020 | −1.016 | −0.719 | −0.967 | −0.908 | −0.974 | −0.514 | −0.512 | −0.512 | −1.004 | −0.823 | −0.822 | −0.830 |
| FR | 0.794 | **0.839** | 0.849 | 0.756 | **0.899** | 0.825 | **0.946** | 0.509 | 0.522 | 0.458 | 0.861 | 0.408 | 0.428 | 0.700 |
| VM | −2.009 | −1.991 | −2.037 | −1.889 | −1.918 | −1.918 | −1.751 | −2.181 | −2.207 | −2.176 | −1.927 | −2.250 | −2.197 | −2.035 |
| RSD | 0.789 | 0.833 | 0.851 | 0.749 | 0.897 | 0.825 | 0.941 | 0.502 | 0.503 | 0.445 | 0.862 | 0.419 | 0.500 | 0.701 |
| SSA (ours) | **0.803** | **0.839** | **0.851** | **0.792** | **0.899** | **0.829** | 0.943 | **0.580** | **0.592** | **0.560** | **0.863** | **0.440** | **0.517** | **0.731** |
| Oracle | 0.856 | 0.890 | 0.889 | 0.862 | 0.917 | 0.873 | 0.960 | 0.635 | 0.652 | 0.635 | 0.895 | 0.519 | 0.671 | 0.789 |

Table 5: Test $R^2$ scores on Biwi Kinect (higher is better). The best scores are **bolded**.

| Method | Female → Male | | | Male → Female | | | Mean |
|---|---|---|---|---|---|---|---|
| | Pitch | Roll | Yaw | Pitch | Roll | Yaw | |
| Source | 0.759 | 0.956 | 0.481 | 0.763 | 0.791 | 0.485 | 0.706 |
| DANN | $0.698_{\pm0.03}$ | $0.826_{\pm0.03}$ | $-0.039_{\pm0.08}$ | $0.711_{\pm0.01}$ | $0.850_{\pm0.01}$ | $0.076_{\pm0.05}$ | $0.520_{\pm0.02}$ |
| TTT | $-0.062_{\pm0.20}$ | $0.606_{\pm0.00}$ | $0.031_{\pm0.02}$ | $0.750_{\pm0.00}$ | $0.725_{\pm0.00}$ | $-0.321_{\pm0.00}$ | $0.288_{\pm0.03}$ |
| BN-adapt | $0.771_{\pm0.00}$ | $0.953_{\pm0.00}$ | $0.493_{\pm0.01}$ | $0.832_{\pm0.00}$ | $0.842_{\pm0.00}$ | $\mathbf{0.585_{\pm0.00}}$ | $0.746_{\pm0.00}$ |
| Prototype | $-318_{\pm0.00}$ | - | - | - | - | - | - |
| FR | $-1.27_{\pm0.70}$ | $0.742_{\pm0.05}$ | $-2.69_{\pm0.79}$ | $0.622_{\pm0.06}$ | $0.855_{\pm0.00}$ | $-0.406_{\pm0.30}$ | $-0.357_{\pm0.23}$ |
| VM | $-0.302_{\pm0.00}$ | $-0.062_{\pm0.00}$ | $-0.089_{\pm0.00}$ | $-0.101_{\pm0.01}$ | $-0.045_{\pm0.00}$ | $0.001_{\pm0.00}$ | $-0.100_{\pm0.00}$ |
| RSD | $0.783_{\pm0.02}$ | $0.954_{\pm0.00}$ | $0.489_{\pm0.02}$ | $0.832_{\pm0.00}$ | $0.846_{\pm0.01}$ | - | - |
| SSA (ours) | $\mathbf{0.860_{\pm0.00}}$ | $\mathbf{0.962_{\pm0.00}}$ | $\mathbf{0.513_{\pm0.01}}$ | $\mathbf{0.869_{\pm0.00}}$ | $\mathbf{0.886_{\pm0.00}}$ | $0.575_{\pm0.00}$ | $\mathbf{0.778_{\pm0.00}}$ |
| Oracle | $0.966_{\pm0.00}$ | $0.981_{\pm0.00}$ | $0.804_{\pm0.00}$ | $0.959_{\pm0.00}$ | $0.970_{\pm0.00}$ | $0.811_{\pm0.00}$ | $0.915_{\pm0.00}$ |

the baselines sometimes even underperformed Source. Moreover, SSA worked well not only on image data but also tabular data.

### 4.3.3 ABLATION STUDY

We performed an ablation study on the subspace detection and dimension weighting. For the SSA variant without subspace detection (*i.e.*, naively aligning the entire feature space), we simply selected the top-$K$ feature dimensions that had the largest variances. In this case, we directly used the weight of the linear regressor $h_\psi$ to compute the dimension weight $\alpha_d$ as $\alpha_d = 1 + |w_d|$ instead of Equation (5). Table 6 shows the test $R^2$ scores with and without subspace detection and dimension weighting on each dataset. Without subspace detection, the scores were worse than Source on MNIST and Biwi Kinect, and of the same level as simple baselines like BN-adapt (Benz et al., 2021) on UTKFace (Table 4). In contrast, subspace detection significantly improved the scores on all three datasets. Dimension weighting also improved the scores, although the gain was smaller than in the case of subspace detection. This is because the variance of the feature subspace dimension correlates with the weight; *i.e.*, the top-$K$ selected dimensions with respect to variance tended to have high importance to the output. Table 7 lists the correlation coefficients between the top $K = 100$ variances of the source features along the source bases $\lambda_d^s$ and the corresponding dimension weight $\alpha_d$. We can see that there are strong correlations in the three datasets we used. But we can

Table 6: Test $R^2$ scores of SSA with and without subspace detection and dimension weighting. Scores averaged over corruption types and gender-task combinations are reported for UTKFace and Biwi Kinect, respectively. The best scores are **bolded**.

| Subspace | Weight | SVHN | UTKFace | Biwi Kinect | California |
|---|---|---|---|---|---|
| | | $0.333_{\pm0.04}$ | $0.642_{\pm0.27}$ | $0.672_{\pm0.24}$ | $0.626_{\pm0.01}$ |
| ✓ | | $0.508_{\pm0.04}$ | $0.728_{\pm0.16}$ | $0.778_{\pm0.17}$ | $0.633_{\pm0.00}$ |
| ✓ | ✓ | $\mathbf{0.511_{\pm0.03}}$ | $\mathbf{0.731_{\pm0.16}}$ | $\mathbf{0.778_{\pm0.17}}$ | $\mathbf{0.639_{\pm0.00}}$ |
| Source | | 0.406 | 0.020 | 0.706 | 0.605 |

Table 7: Correlation coefficients between the top $K = 100$ variances of source features along the source bases $\lambda_d^s$ and weight $\alpha_d$ in Equation (5).

| Dataset | SVHN | UTKFace | Biwi Kinect (Female, Pitch) |
|---|---|---|---|
| Correlation | 0.787 | 0.917 | 0.782 |

| (a) SVHN | (b) UTKFace (Gaussian noise) | (c) Biwi Kinect (Female → Male, Pitch) |
|---|---|---|

Figure 2: Reconstruction error of features reconstructed with the source bases relative to the original target features. Note that Source and Prototype are the same, since Prototype does not update the feature extractor of the model. FR (Eastwood et al., 2022) and VM are not plotted in (a) because they had huge errors.

expect that dimension weighting can raise the importance of the feature dimensions that have low variance but affect the output, which further improves regression performance.

Next, we investigated the effect of the number of feature subspace dimensions $K$. We varied $K$ within $\{10, 25, 50, 75, 100, 200, 400, 1000, 2048\}$. Table 8 shows the test $R^2$ scores. Although the best $K$ differs among the datasets, $K = 100$ consistently produced competitive results. With increasing $K$, the best or competitive scores were when $K$ was close to the number of the subspace dimensions in Table 1. This indicates the importance of the subspace feature alignment. When $K \geq 400$ in MNIST and $K \geq 1000$ in Biwi Kinect, the loss became unstable or diverged because SSA attempted to align too many degenerated feature dimensions. In contrast, although setting $K \geq 1000$ gave the good scores on UTKFace, $K = 100$ produced a competitive score. Appendix D.4 provides the results for California Housing, which we observed the same tendency with the other three datasets. From these results, although $K$ is a hyperparameter, we can determine $K$ before accessing the test dataset by calculating the number of the source feature subspace dimensions.

### 4.3.4 Feature Subspace Analysis

**Feature reconstruction.** To verify that the reason why the baseline methods degrade the regression performance is that they affect the feature subspace learned by the source model as mentioned in Section 4.3.2, we examined the reproducibility of the target features with the source bases $\mathbf{V}^s$ after TTA. That is, the target features can be represented by a linear combination of the source bases if the model retains the source subspace throughout TTA and the target features fit within the subspace. To measure this quantitatively, we computed the reconstruction error $L$ as the Euclidean distance between a target feature vector $\mathbf{z}^t$ and $\mathbf{z}_r^t$, the one reconstructed with $n$ source bases:

Table 8: Test $R^2$ scores of SSA for different numbers of feature subspace dimensions $K$. The best scores are **bolded**.

| $K$ | MNIST | UTKFace | Biwi Kinect |
|---|---|---|---|
| 10 | 0.494 | 0.693 | 0.688 |
| 25 | **0.538** | 0.717 | 0.761 |
| 50 | 0.524 | 0.728 | 0.767 |
| 75 | 0.516 | **0.732** | 0.774 |
| 100 | 0.511 | 0.731 | **0.778** |
| 200 | 0.496 | 0.731 | 0.771 |
| 400 | - | 0.731 | 0.755 |
| 1000 | - | **0.732** | - |
| 2048 | - | 0.725 | - |

$$L = \|\mathbf{z}_r^t - \mathbf{z}^t\|_2, \quad \mathbf{z}_r^t = \boldsymbol{\mu}^s + \sum_{d=1}^{n} ((\mathbf{z}^t - \boldsymbol{\mu}^s)^\top \mathbf{v}_d^s) \mathbf{v}_d^s, \tag{8}$$

where $\mathbf{z}^t$ is a target feature vector extracted with the model after TTA, and $n$ is the number of dimensions of the source subspace listed in Table 1.

Figure 2 plots the reconstruction error $L$ versus $n$ on the three datasets. The error decreased as $n$ increased for all methods, but SSA reduced the error with a smaller $n$ than in those of the baselines, indicating that it could make the target features fit within the source subspace. Especially in the case

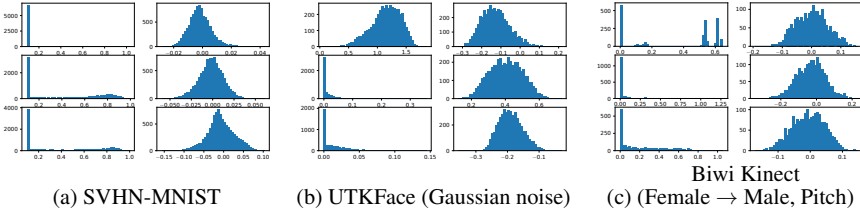

(a) SVHN-MNIST          (b) UTKFace (Gaussian noise)          Biwi Kinect
(c) (Female → Male, Pitch)

Figure 3: Histograms of three randomly selected target feature dimensions. **Left**: Original features. **Right**: Projected features.

of Biwi Kinect (c), the baseline methods produced larger errors than Source; *i.e.*, they broke the learned subspace.

In terms of feature alignment, Figure 4 in Appendix D.2 visualizes and evaluates the feature gap between the domains in the subspace.

**Another effect of subspace detection.** Subspace detection has another effect on feature alignment. We used the KL divergence between two diagonal Gaussian distributions in Equation (3) to measure the distribution gap between the source and target features, under the assumption that the features follow a Gaussian distribution. Here, while it is not clear that the assumption actually holds especially when features are output through activation functions like ReLU as in ResNet, we argue that the subspace detection of SSA has, in addition to an effective feature alignment, the effect of making such features follow a distribution close to a Gaussian.

We visualized the histograms of the target features $\mathbf{z}_i^{\mathrm{t}}$ extracted with the source model and ones projected to the source subspace with Equation (6). Figure 3 show the histograms of three randomly selected dimensions of the original and projected features of the three datasets. In the left column of each figure, the histograms of the original features concentrate on zero because of the ReLU activation and do not follow a Gaussian distribution, which makes the KL divergence computation with Equation (3) inaccurate. On the other hand, the histograms of the projected features in the right columns are close to Gaussians. Thus, subspace detection makes it easier to align the features with the Gaussian KL divergence.

The reason why the projected features follow a Gaussian distribution can be interpreted as follows. From Equation (6), the $k$-th element of a projected feature vector $\tilde{z}_{i,k}^{\mathrm{t}}$ is

$$\tilde{z}_{i,k}^{\mathrm{t}} = \sum_{d=1}^{D} (z_{i,d}^{\mathrm{t}} - \mu_d^{\mathrm{s}}) v_{k,d}^{\mathrm{s}}. \tag{9}$$

Here, we regard each term $a_{i,d} := (z_{i,d}^{\mathrm{t}} - \mu_d^{\mathrm{s}}) v_{k,d}^{\mathrm{s}}$ as a random variable. Assuming that $a_{i,d}$ is independent of the feature dimension $d$, the central limit theorem guarantees that the distribution of the projected features, *i.e.*, the sum of $a_{i,d}$, becomes closer to a Gaussian as the total number of dimensions $D$ increases.

## 5 RELATED WORK

### 5.1 UNSUPERVISED DOMAIN ADAPTATION

Unsupervised domain adaptation (UDA) has been actively studied as a way to transfer knowledge in the source domain to the target domain (Csurka, 2017). Theoretically, it is known that the upper bound of the error on the target domain includes a distribution gap term between the source and target domains (Ben-David et al., 2010; Ganin et al., 2016; Nguyen et al., 2022). For regression, Cortes & Mohri (2011) theoretically explored regression UDA. RSD (Chen et al., 2021) and DARE-GRAM (Nejjar et al., 2023) take into account that the feature scale matters in regression and explicitly align the feature scale during the feature alignment. However, UDA requires the source and target datasets to be accessed simultaneously during training, which can be restrictive when datasets cannot be accessed due to privacy or security concerns, or storage limitations.

More recently, source-free domain adaptation (SFDA), which does not access the source dataset during adaptation, has been studied. The SFDA setting is similar to TTA in that SFDA adapts

models with only unlabeled target data. However, SFDA requires to store the whole target dataset and access the dataset for multiple epochs to train additional models (Li et al., 2020; Xia et al., 2021; Chu et al., 2022; Sanyal et al., 2023; He et al., 2024) or perform clustering (Liang et al., 2020). On the other hand, TTA does not train additional models nor access the target dataset for multiple epochs, which enables instant adaptation with low computational resource and storage.

## 5.2 TEST-TIME TRAINING

Test-time training (TTT) is also similar to TTA as it adapts models with unlabeled target data. The main difference is that TTT requires to modify the model architecture and training procedure in the source domain. The main approach of TTT is additionally training an self-supervised branch simultaneously with the main supervised task in the source domain. Then, during adaptation, the model is updated via minimizing the self-supervised loss on the target data. On the basis of this approach, TTT methods with various self-supervised tasks have been proposed such as rotation prediction (Sun et al., 2020), contrastive learning (Liu et al., 2021), clustering (Hakim et al., 2023), or distribution modeling with normalizing flow (Osowiechi et al., 2023). However, adding additional losses to the training prohibits the use of off-the-shelf pre-trained models or may potentially affect the performance of the main task. In contrast, TTA accepts arbitrary training methods in the source domain and thus off-the-shelf-models can be adapted.

## 5.3 TEST-TIME ADAPTATION

Test-time adaptation (TTA) aims to adapt a model trained on the source domain to the target domain without accessing the source data (Liang et al., 2023). TTA can be applied in a wider range of situations than SFDA and TTT in that TTA does not train additional models or modify the model architecture and source pre-training. TTA for classification has attracted attention for its practicality. Various types of TTA methods have been developed.

**Entropy-based**. Wang et al. (2021) found that the entropy of prediction strongly correlates with accuracy on the target domain and proposed test-time entropy minimization (Tent), which is the most representative of the TTA methods. BACS (Zhou & Levine, 2021), MEMO (Zhang et al., 2022), EATA (Niu et al., 2022) and DELTA (Zhao et al., 2023) follow the idea of Tent and improve adaptation performance. T3A (Iwasawa & Matsuo, 2021) adjusts the prototype in the feature space during testing. IST (Ma, 2024) employs graph-based pseudo label modification. However, these TTA methods are designed for classification and cannot be applied to regression. For instance, computing entropy, which is widely adopted in TTA, requires a predictive probability for each class, whereas ordinary regression models only output a single predicted value. Thus, we investigate an approach that does not rely on entropy.

**Feature alignment**. Feature alignment is based on the insight of UDA and makes the target feature distribution close to the source one. Since accessing the source data is restricted in the TTA setting, methods based on the feature alignment match the statistics of the target features to those of the pre-computed source. BN-adapt (Benz et al., 2021) updates the feature mean and variance stored in batch normalization (BN) layers (Ioffe & Szegedy, 2015). DELTA (Zhao et al., 2023) modifies BN and introduces class-wise loss re-weighting. CFA (Kojima et al., 2022), BUFR (Eastwood et al., 2022), CAFe (Adachi et al., 2023), and CAFA (Jung et al., 2023) incorporate pre-computed source statistics. Although some of these methods are directly applicable to regression, we have observed that they are not effective or even degrade regression performance.

**Other tasks**. TTA for depth estimation (Li et al., 2023) super resolution (Deng et al., 2024), point cloud (Wang et al., 2024), and person re-identification (Adachi et al., 2024) are proposed. But they have task-specific architectures or methods and cannot be applied to ordinary regression.

## 6 CONCLUSION

We proposed significant-subspace alignment (SSA), a novel test-time adaptation method for regression models. Since we have found that the naive feature alignment fails in regression TTA because of the learned features being distributed in a small subspace, we incorporated subspace detection and dimension weighting procedures into SSA. Experimental results show that SSA achieved higher $R^2$ scores on various regression tasks than did baselines that were originally designed for classification tasks. We will extend TTA to further broader tasks and settings such as concept drift in the future.

**Ethics statement.** The potential ethical concern is that a model may have fairness or bias issues in certain sensitive applications if the model adapts to biased target data. The model's behavior should be carefully monitored in such a situation.

**Reproducibility statement.** Details on the datasets and experimental settings are described in Sections 4.1 and 4.2 and Appendix C. We also provide the code in the supplementary material.

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

APPENDIX

## A  LIMITATION

One limitation of SSA is that it assumes a covariate shift, where $p(y|\mathbf{x})$ does not change. Addressing distribution shifts where $p(y|\mathbf{x})$ changes, *e.g.*, concept drift, will specifically be addressed as a target in future work.

## B  EVALUATION METRIC

We used the $R^2$ score (coefficient of determination) to measure the performance of the regression models. $R^2$ is computed as follows:

$$R^2 = 1 - \frac{\sum_{i=1}^{N}(\hat{y}_i - y_i)^2}{\sum_{i=1}^{N}(y_i - \bar{y})^2}, \tag{10}$$

where $\hat{y}_i$ is a predicted value of the regression model, $y_i$ is the ground-truth, and $\bar{y} = (1/N)\sum_{i=1}^{N} y_i$ is the mean of the ground truth values. $R^2$ is close to 1 when the regression model is accurate.

## C  EXPERIMENTAL SETTINGS

We used PyTorch (Paszke et al., 2019) and PyTorch-Ignite (Fomin et al., 2020) to make the implementations of the source pre-training, proposed method, and baselines. We conducted the experiments with a single NVIDIA A100 GPU.

### C.1  DATASETS

**SVHN** (Netzer et al., 2011): We downloaded SVHN via `torchvision.datasets.SVHN`. It can be used for non-commercial purposes only[1].

**MNIST** (LeCun et al., 1998b): We downloaded MNIST via `torchvison.datasets.MNIST`. We could not find any license information for MNIST.

**UTKFace** (Zhang et al., 2017): We downloaded UTKFace via the official site[2]. It is available for non-commercial research purposes only.

**Biwi Kinect** (Fanelli et al., 2013): We downloaded Biwi Kinect via Kaggle[3]. It is released under Attribution-NonCommercial-NoDerivatives 4.0 International (CC BY-NC-ND 4.0) license.

**California Housing** (Nugent, 2017): We downloaded the dataset from Kaggle[4]. It is released in public domain.

### C.2  PRE-TRAINING ON THE SOURCE DOMAIN

**SVHN**: We trained ResNet-26 (He et al., 2016) from scratch. For optimization, we used Adam (Kingma & Ba, 2015) and set the learning rate to 0.0001, weight decay to 0.0005, batch size to 64, and number of epochs to 100. We minimized the MSE loss between the predicted values and digit labels. For the implementation of ResNet-26, we used the PyTorch Image Models (timm) library (Wightman, 2019).

**UTKFace**: We randomly split the dataset into 80% for training and 20% for validation. We fine-tuned ResNet-50 pre-trained on ImageNet (Deng et al., 2009). We minimized the MSE loss to predict the ages of the persons in the images. For optimization, we used the same hyperparameters

---

[1] `http://ufldl.stanford.edu/housenumbers/`
[2] `https://susanqq.github.io/UTKFace/`
[3] `https://www.kaggle.com/datasets/kmader/biwi-kinect-head-pose-database`
[4] `https://www.kaggle.com/datasets/camnugent/california-housing-prices`

as the above SVHN case. For the implementation of ResNet-50, we used torchvision (maintainers & contributors, 2016) with IMAGENET1K_V2 initial weights.

**Biwi Kinect**: We split the dataset into male and female images and further randomly split them into 80% for training and 20% for validation. We fine-tuned ResNet-50 pre-trained on ImageNet in the same way as the UTKFace case. We separately trained the models to predict each of the three head angles, *i.e.*, pitch, roll, and yaw, of the persons in the images. In total, we pre-trained six source models ({male, female} × {pitch, yaw, roll}).

**California Housing**: We extracted the data of non-coastal areas for the source domain and split them into 90% for training and 10% for validation. We standardized the whole dataset using the source mean and standard deviation. We trained a five-layer MLP, with 100-dimensional hidden layers, ReLU activation, and batch normalization.

## C.3 TEST-TIME ADAPTATION

As for setting the hyperparameters of the baseline methods, we basically followed their original papers. For adaptation, we adopted an offline manner for fair comparison, *i.e.*, we ran TTA for one epoch and then ran evaluation, which is widely adopted in existing TTA works (Wang et al., 2021; Zhou & Levine, 2021; Eastwood et al., 2022). For the evaluation, we ran each TTA three times with different random seeds and reported the means of the scores.

**Source**: We simply fixed the source-pretrained model (*i.e.*, model.eval() in PyTorch) and performed inference.

**BN-adapt** (Benz et al., 2021): updates the feature mean and variance stored in the batch normalization layers during testing, *i.e.*, ran inference with model.train() mode in PyTorch.

**Feature restoration (FR)** (Eastwood et al., 2022): uses the source statistics of the features and outputs as a form of dimension-wise histogram and aligns the target feature histogram to the source one. The original FR uses the histograms of the features and logits pre-computed with the source dataset. Since our focus is on regression models, we used the outputs instead of logits. We set the number of bins of the histograms to eight and the temperature $\tau$ of soft-binning to 0.01 following Eastwood et al. (2022). We set learning rate to 0.0001 (this value gave the best score).

**Prototype**: We tweaked T3A (Iwasawa & Matsuo, 2021), which regards the weights of the last fully-connected layer as the prototype of each class and updates the prototypes with the mean of the arriving target features during testing. We regarded $\mathbf{w}$ of the linear regressor $h_\psi$ as a single prototype and updated it with the mean of the target features. Although T3A determines whether to use a feature for making an update by using entropy, we omitted this component since we cannot compute entropy with regression models.

**Variance minimization (VM)**: makes augmented views of input images and minimizes output variance. This is a modification of MEMO (Zhang et al., 2022), which minimizes the marginal entropy of the augmented views of inputs in the same manner. We set the number of augmented views to 32 per input and set the learning rate to 0.001 with Adam optimizer.

**Representation subspace distance (RSD)** (Chen et al., 2021): The original RSD is a UDA method designed for regression, which aligns the SVD bases between source and target mini-batches. To adapt to TTA, we pre-computed the SVD of the source features wit the whole source dataset instead of mini-batches before TTA and align the target feature SVD with the RSD loss during testing.

**SSA (ours)**: Algorithm 1 lists the procedure of SSA. For optimization, we used Adam (Kingma & Ba, 2015) with a learning rate= 0.001, $(\beta_1, \beta_2) = (0.9, 0.999)$, and weight decay= 0, which is the default setting in PyTorch (Paszke et al., 2019). We set the batch size to 64 following other TTA baselines.

**Test-time training (TTT)** (Sun et al., 2020): incorporates a self-supervised rotation prediction task during pre-training in the source domain; then it updates the feature extractor by minimizing the self-supervised loss during testing. The rotation-prediction branch is a linear layer that takes a feature $\mathbf{z}$ and outputs four logits corresponding to the rotation angles $\{0°, 90°, 180°, 270°\}$ of an input image. For optimization, we used SGD and set the learning rate to 0.001 in accordance with Sun et al. (2020).

---

**Algorithm 1** Significant-subspace alignment (SSA).

---

**Input:** Pre-trained source model $f_\theta$, source bases $\mathbf{V}^\mathrm{s}$, source mean $\boldsymbol{\mu}^\mathrm{s}$, source variances $\boldsymbol{\lambda}^\mathrm{s}$, target dataset $\mathcal{T}$
**Output:** Adapted model $f_{\theta'}$
   Compute weights for each dimension of the source subspace $\alpha_d$ according to Equation (5)
   **for all** mini-batch $\{\mathbf{x}_i^\mathrm{t}\}_i$ in $\mathcal{T}$ **do**
     Extract target features $\{\mathbf{z}_i^\mathrm{t} = g_\phi(\mathbf{x}_i^\mathrm{t})\}_i$
     Project target features $\{\mathbf{z}_i^\mathrm{t}\}_i$ into $\{\tilde{\mathbf{z}}_i^\mathrm{t}\}_i$ according to Equation (6)
     Compute projected target mean $\tilde{\boldsymbol{\mu}}^\mathrm{t}$ and variances $\tilde{\boldsymbol{\sigma}}^{\mathrm{t}\,2}$ analogously to Equation (1)
     Update $\phi$ of the feature extractor $g_\phi$ to minimize $\mathcal{L}_\mathrm{TTA}$ according to Equation (7)
   **end for**

---

Table 9: Comparison between using KL divergence and 2-Wasserstein distance for feature alignment in SSA on SVHN-MNIST. The top row is our method.

| Metric | Subspace detection | $R^2$ ($\uparrow$) | RMSE ($\downarrow$) |
|--------|:------:|:------:|:------:|
| KL | ✓ | $\mathbf{0.511}_{\pm\mathbf{0.03}}$ | $\mathbf{2.024}_{\pm\mathbf{0.06}}$ |
| KL | | $0.338_{\pm 0.04}$ | $2.355_{\pm 0.07}$ |
| 2WD | ✓ | $0.425_{\pm 0.02}$ | $2.196_{\pm 0.04}$ |
| 2WD | | $0.342_{\pm 0.04}$ | $2.348_{\pm 0.07}$ |
| L1 | ✓ | $0.472_{\pm 0.03}$ | $2.104_{\pm 0.05}$ |
| L1 | | $0.347_{\pm 0.06}$ | $2.337_{\pm 0.11}$ |
| | Source | $0.406$ | $2.232$ |

**Domain adversarial neural network (DANN)** (Ganin et al., 2016): is an unsupervised domain adaptation method which adversarially trains a feature extractor and domain discriminator to learn domain-invariant features. We trained the domain discriminator during training in addition to the main regression model. We used layer4 of ResNet for the discriminator. We scheduled the learning rate and the weight of the domain adaptation loss by following Ganin et al. (2016).

**Oracle**: fine-tunes the model using labels during testing, *i.e.*, the upper bound of the performance.

## D   ADDITIONAL EXPERIMENTAL RESULTS

### D.1   METRIC FOR FEATURE ALIGNMENT

We used the KL divergence between two Gaussian distributions in Equation (3) for the feature alignment in SSA. One may suppose that other metrics could also be used, since the variance term in the denominator makes the naive TTA loss in Equation (2) unstable, as mentioned in Section 3.1. Here, we tried the 2-Wasserstein distance (2WD) between two Gaussian distributions (Dowson & Landau, 1982) and the L1 norm of the statistics:

$$W_2^2\left(\mathcal{N}(\mu_1, \sigma_1^2), \mathcal{N}(\mu_2, \sigma_2^2)\right) = (\mu_1 - \mu_2)^2 + (\sigma_1 - \sigma_2)^2, \tag{11}$$

$$L_1\left(\mathcal{N}(\mu_1, \sigma_1^2), \mathcal{N}(\mu_2, \sigma_2^2)\right) = |\mu_1 - \mu_2| + |\sigma_1 - \sigma_2|. \tag{12}$$

We replaced the KL divergence with the 2WD and L1 in Equation (7) as follows:

$$\mathcal{L}_\mathrm{TTA\text{-}2WD} = \sum_{d=1}^{K} \alpha_d W_2^2\left(\mathcal{N}(\tilde{\mu}_d^\mathrm{t}, \tilde{\sigma}_d^{\mathrm{t}\,2}), \mathcal{N}(0, \lambda_d^\mathrm{s})\right) = \sum_{d=1}^{K} \alpha_d \left\{ (\tilde{\mu}_d^\mathrm{t})^2 + \left(\sqrt{\tilde{\sigma}_d^{\mathrm{t}\,2}} - \sqrt{\lambda_d^\mathrm{s}}\right)^2 \right\}, \tag{13}$$

$$\mathcal{L}_\mathrm{TTA\text{-}L1} = \sum_{d=1}^{K} \alpha_d L_1\left(\mathcal{N}(\tilde{\mu}_d^\mathrm{t}, \tilde{\sigma}_d^{\mathrm{t}\,2}), \mathcal{N}(0, \lambda_d^\mathrm{s})\right) = \sum_{d=1}^{K} \alpha_d \left\{ |\tilde{\mu}_d^\mathrm{t}| + \left|\sqrt{\tilde{\sigma}_d^{\mathrm{t}\,2}} - \sqrt{\lambda_d^\mathrm{s}}\right| \right\}. \tag{14}$$

Tables 9 to 11 compare the effects of using the KL divergence and 2WD on SVHN-MNIST, UTK-Face, and Biwi Kinect. The KL divergence with subspace detection (SSA) achieved highest $R^2$

Table 10: Test $R^2$ scores of cases using KL divergence and 2-Wasserstein distance for feature alignment in SSA on UTKFace. The top row is our method.

| Metric | Subspace detection | Defocus blur | Motion blur | Zoom blur | Contrast | Elastic transform | Jpeg compression | Pixelate | Gaussian noise | Impulse noise | Shot noise | Brightness | Fog | Snow | Mean |
|---|---|---|---|---|---|---|---|---|---|---|---|---|---|---|---|
| KL | ✓ | 0.803 | 0.839 | **0.851** | **0.792** | 0.899 | 0.829 | 0.943 | **0.580** | **0.592** | **0.560** | **0.863** | **0.440** | **0.517** | **0.731** |
| KL | | 0.826 | 0.853 | 0.825 | 0.752 | 0.904 | 0.843 | 0.944 | 0.377 | 0.421 | 0.294 | 0.842 | 0.246 | 0.205 | 0.641 |
| 2WD | ✓ | 0.816 | 0.843 | 0.826 | 0.729 | 0.903 | 0.830 | 0.946 | 0.353 | 0.424 | 0.302 | 0.824 | 0.177 | 0.192 | 0.628 |
| 2WD | | 0.827 | **0.854** | 0.832 | 0.755 | 0.906 | 0.845 | 0.946 | 0.389 | 0.439 | 0.298 | 0.846 | 0.223 | 0.238 | 0.646 |
| L1 | ✓ | **0.834** | **0.858** | **0.851** | 0.775 | **0.910** | **0.849** | **0.949** | 0.484 | 0.546 | 0.489 | 0.845 | 0.334 | 0.206 | 0.687 |
| L1 | | 0.830 | 0.854 | 0.840 | 0.744 | 0.905 | 0.847 | 0.942 | 0.362 | 0.418 | 0.288 | 0.848 | 0.233 | 0.277 | 0.645 |
| Source | | 0.410 | 0.159 | 0.658 | −3.906 | 0.711 | 0.069 | 0.595 | −2.536 | −2.539 | −2.522 | 0.661 | −0.029 | −0.544 | −0.678 |

Table 11: Test $R^2$ scores of cases using KL divergence and 2-Wasserstein distance for feature alignment in SSA on Biwi Kinect. The top row is our method.

| Metric | Subspace detection | Female → Male | | | Male → Female | | | |
|---|---|---|---|---|---|---|---|---|
| | | Pitch | Roll | Yaw | Pitch | Yaw | Roll | Mean |
| KL | ✓ | $\mathbf{0.860_{\pm 0.00}}$ | $\mathbf{0.962_{\pm 0.00}}$ | $\mathbf{0.513_{\pm 0.01}}$ | $\mathbf{0.869_{\pm 0.00}}$ | $0.886_{\pm 0.00}$ | $0.575_{\pm 0.00}$ | $\mathbf{0.778_{\pm 0.00}}$ |
| KL | | $0.525_{\pm 0.04}$ | $0.945_{\pm 0.00}$ | $0.240_{\pm 0.03}$ | $0.835_{\pm 0.01}$ | $0.874_{\pm 0.01}$ | $0.613_{\pm 0.01}$ | $0.672_{\pm 0.01}$ |
| 2WD | ✓ | $0.708_{\pm 0.05}$ | $0.954_{\pm 0.00}$ | $0.465_{\pm 0.01}$ | $0.765_{\pm 0.01}$ | $0.916_{\pm 0.01}$ | $0.617_{\pm 0.00}$ | $0.738_{\pm 0.01}$ |
| 2WD | | $0.540_{\pm 0.05}$ | $0.949_{\pm 0.00}$ | $0.279_{\pm 0.02}$ | $0.829_{\pm 0.01}$ | $0.862_{\pm 0.02}$ | $0.598_{\pm 0.01}$ | $0.676_{\pm 0.01}$ |
| L1 | ✓ | $0.750_{\pm 0.07}$ | $0.958_{\pm 0.00}$ | $0.482_{\pm 0.01}$ | $0.858_{\pm 0.00}$ | $\mathbf{0.922_{\pm 0.00}}$ | $\mathbf{0.641_{\pm 0.00}}$ | $0.768_{\pm 0.01}$ |
| L1 | | $0.562_{\pm 0.03}$ | $0.949_{\pm 0.00}$ | $0.314_{\pm 0.04}$ | $0.802_{\pm 0.00}$ | $0.861_{\pm 0.01}$ | $0.613_{\pm 0.00}$ | $0.684_{\pm 0.01}$ |
| Source | | 0.759 | 0.956 | 0.481 | 0.763 | 0.791 | 0.485 | 0.706 |

scores in almost all cases. In contrast, the 2WD variant of SSA produced only a slight improvement over Source on SVHN-MNIST (Table 9) and sometimes it had even worse scores than Source on Biwi Kinect (Table 11). This degradation of 2WD is because the scale of the variance $\sigma_d^2$ is different among feature dimensions $d$. The KL divergence can absorb the difference in scale since it includes the ratio of the variances, as in Equation (7).

## D.2 FEATURE VISUALIZATION

Figure 4 illustrates PCA visualizations of the source and target features after TTA. In the visualizations of SVHN-MNIST (a) and Biwi Kinect (c), we can see that SSA makes the target feature distribution fit within the source distribution while the target features of the baselines protrude from the source distribution. In UTKFace (b), the target features of Source significantly degenerate to a single point, but the other methods alleviate this. In Figure 4, we also report the optimal transport distance (OTD) between the features of the both domains, which evaluates the feature alignment quantitatively. In addition to that our SSA alleviates the feature distribution gap in the subspace, SSA retains the source subspace better, as shown in Figure 2 of Section 4.3.4.

We also visualized the source and target features with UMAP (McInnes et al., 2018) in Figure 5 to see the relation not limited within the top-2 principal components. We first trained the UMAP mapping with the source features projected onto the top $K = 100$ dimensional principal component space before TTA, which are shared among TTA methods. Then, we mapped the target features after TTA with the learned UMAP mapping. We can also see that the target feature distribution becomes closer to the source one by our SSA.

## D.3 EFFECT OF ORIGINAL FEATURE DIMENSIONS ON THE SUBSPACE

As mentioned in Section 3.1, many of the feature dimensions have a small effect on the subspace, which makes the naive feature alignment ineffective. To verify the effect of changing the original features on the subspace, we computed the gradient of a subspace feature vector $\tilde{\mathbf{z}}$ with respect to

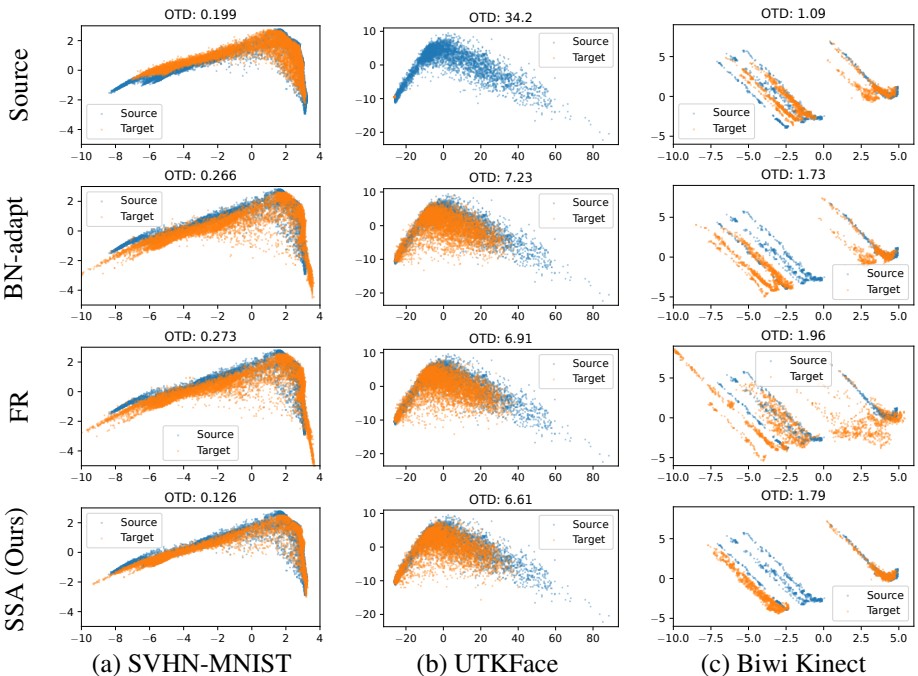

Figure 4: PCA visualizations of source and target features of each dataset and method. The blue and orange dots represent the source and target features, respectively. We also report the optimal transport distance (OTD) between the principal components of the source and target features.

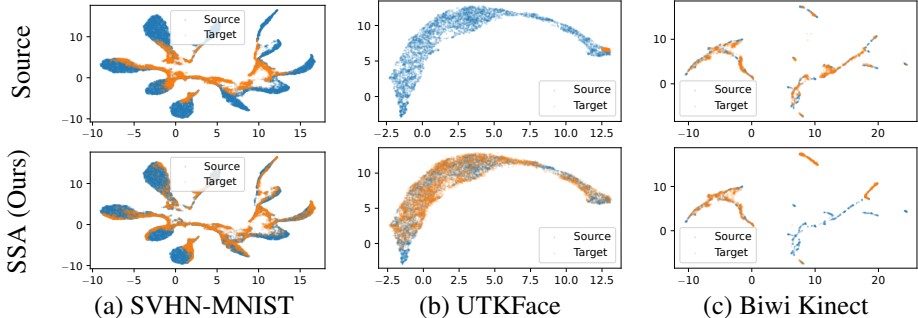

Figure 5: UMAP (McInnes et al., 2018) visualizations of source and target features on each dataset. The blue and orange dots represent the source and target features, respectively.

the $d$-th dimension of the original feature $z_d$. With Equation (6), the norm of the gradient $s_d$ is:

$$s_d = \left\| \frac{\partial \tilde{\mathbf{z}}}{\partial z_d} \right\|_2 = \|(\mathbf{V}^{\mathrm{s}\top})_d\|_2 = \|[v^{\mathrm{s}}_{1,d}, \ldots, v^{\mathrm{s}}_{K,d}]\|_2, \tag{15}$$

which is the norm of the $d$-th row of $\mathbf{V}^{\mathrm{s}}$.

Figure 6 shows the histograms of $s_d$ computed with the three datasets. As expected, most of the dimensions of the original feature space had small effects; only a few dimensions had significant effect to the subspace. Specifically, SVHN-MNIST and Biwi Kinect strongly showed this tendency. In contrast, a larger number of raw feature dimensions affected the subspace in UTKFace compared with the other datasets. This is why the test $R^2$ score was improved from Source without subspace detection in UTKFace, while the scores on the other datasets were lower than those of Source, as shown in Table 6.

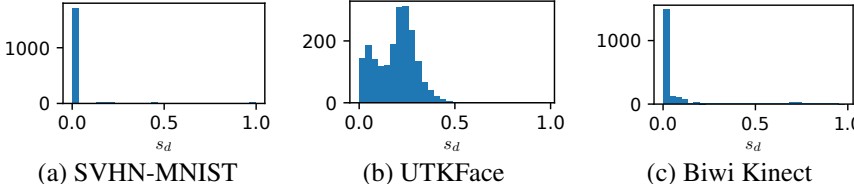

(a) SVHN-MNIST    (b) UTKFace    (c) Biwi Kinect

Figure 6: Histograms of the gradient of a projected feature $\tilde{z}$ with respect to each original feature dimension $z_d$ computed with Equation (15).

Table 12: Test $R^2$ scores of SSA for different numbers of feature subspace dimensions $K$ on California Housing.

| $K$ | 1 | 5 | 10 | 25 | 50 | 75 | 100 |
|-----|-----|-----|-----|-----|-----|-----|-----|
| $R^2$ | 0.496 | 0.625 | **0.639** | 0.625 | - | - | - |

## D.4 ADDITIONAL ABLATION

Table 12 shows the $R^2$ scores on California Housing with changing $K$ of SSA (the same experiment as Table 8). $K = 10$ produced the best result, but $K = 5$ and 25 also gave competitive scores. When $K \geq 50$, which exceeds the number of the subspace dimensions in Table 1, the loss became unstable or diverged because SSA attempted to align too many degenerated feature dimensions, as mentioned in Section 4.3.3.

Tables 13 and 14 shows the test $R^2$ scores without subspace detection (*i.e.*, naively aligning raw features) with respect to $K$. Without subspace detection, the performance of regression on the target domain is limited even when all dimensions of the original feature space are aligned. This result supports the importance of aligning the representative feature subspace with subspace detection.

## D.5 EXPERIMENTS ON VISION TRANSFORMER

We experimented TTA with vision transformer (ViT) (Dosovitskiy et al., 2021) on SVHN-MNIST.

First, we trained a ViT-B/16 regressor on SVHN and computed the number of feature dimensions as Section 4.3.4. We used the output of the penultimate layer for the features.

Table 15 shows the numbers of the valid (zero variance) and subspace dimensions. Since no activation function is applied to the feature vectors of ViTs unlike ResNets, all 768 feature dimensions are valid (non-zero variance). However, the number of the subspace dimensions is 503 in the regression model, which is smaller that of the classification model. Although the subspace dimension is larger than the ResNet cases in Table 1, we can also see the same tendency.

Table 16 shows the TTA results on SVHN-MNIST. SSA outperforms the baselines.

## D.6 MULTI-TASK REGRESSION MODEL

We applied TTA methods to multi-task regression models, which output multiple prediction values at the same time. We trained ResNet-50 regression models on Biwi Kinect. Unlike the experiments on the same dataset in Section 4.3, a single regression model outputs pitch, roll, and yaw angles.

Table 13: Test $R^2$ scores of SSA without subspace detection. The right column represents the score with subspace detection.

| $K$ | 1 | 5 | 10 | 25 | 50 | 75 | 100 | 10 (ours) |
|-----|-----|-----|-----|-----|-----|-----|-----|-----------|
| $R^2$ | 0.524 | 0.605 | 0.625 | 0.620 | 0.584 | - | - | **0.639** |

Table 14: Test $R^2$ scores of SSA without subspace detection. The bottom row represents the scores with subspace detection of $K = 100$ from Table 8.

| $K$ | SVHN | UTKFace | Biwi Kinect |
|---|---|---|---|
| 10 | 0.302 | 0.674 | 0.737 |
| 25 | 0.348 | 0.656 | 0.674 |
| 50 | 0.334 | 0.651 | 0.691 |
| 75 | 0.329 | 0.645 | 0.578 |
| 100 | 0.338 | 0.641 | 0.672 |
| 200 | 0.343 | 0.643 | 0.670 |
| 400 | - | 0.643 | 0.709 |
| 1000 | - | 0.656 | 0.755 |
| 2048 | - | 0.706 | 0.753 |
| 100 (ours) | **0.511** | **0.731** | **0.778** |

Table 15: Number of valid (having non-zero variance) feature dimensions and feature subspace dimensions of the ViT trained on SVHN.

| Regression | | Classification | |
|---|---|---|---|
| #Valid dims | #Subspace dims. | #Valid dims | #Subspace dims. |
| 768 | 503 | 768 | 544 |

Table 17 shows the test $R^2$ score on each target angle. We compared TTA methods that can be easily extended to multi-task regression tasks. Our SSA outperformed the other baselines in most cases. The numbers of the source feature subspace dimensions were 134 and 132 for the male and female models, which are larger than single-task models but much smaller than the entire feature dimensions (2048). Thus, SSA is also effective for multi-task regression models.

### D.7 COMBINATION WITH CLASSIFICATION TTA

Although our SSA is designed for regression, the subspace detection and feature alignment loss can be used to classification since they operates in the feature space and works regardless of model output forms. Here, we applied the subspace detection and feature alignment loss to classification models. We trained ResNet-26 on SVHN and CIFAR10 (Krizhevsky et al., 2009), and tested on MNIST and CIFAR10-C (Hendrycks & Dietterich, 2019).

Table 18 shows the number of the valid feature dimensions and subspace dimensions in classification and regression models on the source datasets. Although the number of the feature dimensions of the classification models are larger than those of the regression models, the subspace is smaller than the entire feature space. Thus, we expect that our SSA is also effective on classification models.

Tables 19 and 20 show the classification accuracies. We can easily combine our method with entropy-based TTA methods in classification. Although SSA significantly improves the accuracy, combining SSA with Tent (Wang et al., 2021) further boosts the accuracy.

Table 16: Test $R^2$ score and RMSE on SVHN-MNIST with ViT.

| Method | $R^2$ ($\uparrow$) | RMSE ($\downarrow$) |
|---|---|---|
| Source | 0.658 | 1.693 |
| Prototype | $0.468_{\pm 0.00}$ | $2.111_{\pm 0.00}$ |
| FR | $0.724_{\pm 0.00}$ | $1.522_{\pm 0.01}$ |
| VM | $-1009_{\pm 11.8}$ | $92.02_{\pm 0.54}$ |
| SSA (ours) | $\mathbf{0.741_{\pm 0.03}}$ | $\mathbf{1.471_{\pm 0.08}}$ |
| Oracle | $0.960_{\pm 0.00}$ | $0.575_{\pm 0.02}$ |

Table 17: Test $R^2$ scores on multi-task Biwi Kinect (higher is better). The best scores are in **bold-face**.

| | Female $\rightarrow$ Male | | | Male $\rightarrow$ Female | | | |
| Method | Pitch | Yaw | Roll | Pitch | Yaw | Roll | Mean |
|---|---|---|---|---|---|---|---|
| Source | 0.904 | **0.628** | 0.960 | 0.794 | 0.419 | 0.854 | 0.760 |
| BN-adapt | $0.912_{\pm 0.00}$ | $0.606_{\pm 0.00}$ | $0.967_{\pm 0.00}$ | $0.791_{\pm 0.00}$ | $0.471_{\pm 0.00}$ | $0.921_{\pm 0.00}$ | 0.778 |
| VM | $-1.875_{\pm 0.37}$ | $-1.962_{\pm 0.73}$ | $-0.093_{\pm 0.03}$ | $-0.365_{\pm 0.08}$ | $-0.143_{\pm 0.12}$ | $-0.000_{\pm 0.02}$ | $-0.740$ |
| RSD | $0.912_{\pm 0.00}$ | $0.606_{\pm 0.00}$ | $0.967_{\pm 0.00}$ | $0.792_{\pm 0.00}$ | $0.472_{\pm 0.00}$ | $0.921_{\pm 0.00}$ | 0.778 |
| SSA (ours) | $\mathbf{0.913_{\pm 0.00}}$ | $0.555_{\pm 0.01}$ | $\mathbf{0.970_{\pm 0.00}}$ | $\mathbf{0.837_{\pm 0.00}}$ | $\mathbf{0.540_{\pm 0.00}}$ | $\mathbf{0.942_{\pm 0.00}}$ | **0.793** |
| Oracle | $0.976_{\pm 0.00}$ | $0.859_{\pm 0.00}$ | $0.988_{\pm 0.00}$ | $0.958_{\pm 0.00}$ | $0.804_{\pm 0.00}$ | $0.979_{\pm 0.00}$ | 0.928 |

Table 18: Comparison of the numbers of valid (having non-zero variance) feature dimensions and feature subspace dimensions (*i.e.*, the rank of the feature covariance matrix) between classification and regression. When training classification models, we discretized the labels.

| | Classification | | Regression | |
| Dataset | #Valid dims | #Subspace dims. | #Valid dims | #Subspace dims. |
|---|---|---|---|---|
| SVHN | 1946 | 64 | 353 | 14 |
| CIFAR10 | 1521 | 86 | 561 | 50 |
| UTKFace | 2048 | 1471 | 2041 | 76 |
| Biwi Kinect (mean) | 2048 | 277 | 713 | 34.5 |
| California Housing (100 dims.) | 100 | 100 | 45 | 40 |

### D.8 HYPERPARAMETER SENSITIVITY

We investigated SSA's sensitivity to the other hyperparameters. Tables 21 to 23 and Tables 24 to 26 show the results when varying the learning rate and batch size, respectively.

Typical ranges of the learning rate and batch size produce competitive performance. For the batch size, a larger batch size results in better performance since the estimation of feature mean and variance becomes more accurate. But batch sizes $\geq 16$ produce competitive performance.

### D.9 ADDITIONAL RESULTS

Tables 27 and 28 provide the performance measured by MAE on UTKFace and Biwi Kinect, which are corresponding to Tables 4 and 5.

### D.10 ONLINE SETUP

We evaluated regression TTA in an batched online setting, where model update and evaluation are performed alternatively with a batch in every iteration. Tables 29 to 31 display the results. Our SSA can outperform the baselines also in the batched online setting.

Table 19: Test classification accuracy (%) on SVHN-MNIST.

| Method | Accuracy |
|---|---|
| Source | 53.82 |
| Tent | $75.68_{\pm 8.49}$ |
| SSA | $79.97_{\pm 0.67}$ |
| Tent+SSA | $\mathbf{80.69_{\pm 0.52}}$ |
| Oracle | $97.48_{\pm 0.02}$ |

Table 20: Test classification accuracy (%) on CIFAR10-C. The best and second scores are in **bold-face** and underlined.

| Method | Brightness | Contrast | Defocus blur | Elastic transform | Fog | Gaussian noise | Impulse noise | Jpeg compression | Motion blur | Pixelate | Shot noise | Snow | Zoom blur | Mean |
|---|---|---|---|---|---|---|---|---|---|---|---|---|---|---|
| Source | 77.80 | 19.97 | 57.74 | 72.33 | 47.62 | 69.06 | 46.97 | **79.82** | 58.92 | 76.88 | 70.44 | 72.18 | 60.75 | 62.34 |
| Tent | 79.06 | 56.88 | 77.00 | 73.70 | 67.41 | 76.98 | 70.02 | 79.22 | 74.28 | 77.80 | 77.70 | 75.27 | 77.08 | 74.03 |
| SSA | 81.34 | 64.98 | 79.11 | 74.48 | 71.89 | **78.44** | 70.48 | 79.63 | 76.03 | 79.48 | 78.91 | **76.55** | 77.80 | 76.09 |
| Tent+SSA | **81.43** | **65.26** | **79.30** | **74.61** | **71.98** | 78.41 | **70.59** | 79.57 | **76.13** | **79.51** | **78.96** | 76.50 | **77.84** | **76.16** |
| Oracle | 85.62 | 77.57 | 83.93 | 79.59 | 78.68 | 83.15 | 76.42 | 84.09 | 81.45 | 84.27 | 83.60 | 81.94 | 83.28 | 81.82 |

Table 21: Test $R^2$ scores of SSA on SVHN-MNIST with different learning rates. Higher is better.

| Learning rate | 0.0001 | 0.0005 | 0.001 | 0.005 | 0.01 |
|---|---|---|---|---|---|
| $R^2$ | $0.472_{\pm0.00}$ | $0.516_{\pm0.01}$ | $0.509_{\pm0.03}$ | $0.510_{\pm0.07}$ | $0.212_{\pm0.21}$ |

Table 22: Test $R^2$ scores of SSA on UTKFace with different learning rates. Higher is better.

| Learning rate | Defocus blur | Motion blur | Zoom blur | Contrast | Elastic transform | Jpeg compression | Pixelate | Gaussian noise | Impulse noise | Shot noise | Brightness | Fog | Snow | Mean |
|---|---|---|---|---|---|---|---|---|---|---|---|---|---|---|
| 0.0001 | 0.781 | 0.828 | 0.836 | 0.755 | 0.895 | 0.820 | 0.941 | 0.526 | 0.534 | 0.469 | 0.860 | 0.420 | 0.474 | 0.703 |
| 0.0005 | 0.786 | 0.828 | 0.834 | 0.767 | 0.893 | 0.820 | 0.938 | 0.551 | 0.566 | 0.513 | 0.861 | 0.434 | 0.508 | 0.715 |
| 0.001 | 0.791 | 0.830 | 0.835 | 0.769 | 0.891 | 0.822 | 0.936 | 0.569 | 0.585 | 0.540 | 0.861 | 0.446 | 0.510 | 0.722 |
| 0.005 | 0.796 | 0.832 | 0.815 | 0.723 | 0.877 | 0.815 | 0.918 | 0.587 | 0.594 | 0.550 | 0.844 | 0.424 | 0.406 | 0.706 |
| 0.01 | 0.769 | 0.804 | 0.769 | 0.668 | 0.863 | 0.792 | 0.899 | 0.523 | 0.574 | 0.464 | 0.811 | 0.327 | 0.293 | 0.658 |

Table 23: Test $R^2$ scores of SSA on Biwi Kinect with different learning rates. Higher is better.

| Learning rate | Female → Male | | | Male → Female | | | Mean |
|---|---|---|---|---|---|---|---|
| | Pitch | Roll | Yaw | Pitch | Roll | Yaw | |
| 0.0001 | 0.787 | 0.955 | 0.505 | 0.842 | 0.849 | 0.581 | 0.753 |
| 0.0005 | 0.840 | 0.958 | 0.519 | 0.861 | 0.875 | 0.577 | 0.772 |
| 0.001 | 0.859 | 0.962 | 0.515 | 0.869 | 0.889 | 0.571 | 0.777 |
| 0.005 | 0.877 | 0.963 | 0.492 | 0.859 | 0.893 | 0.549 | 0.772 |
| 0.01 | 0.879 | 0.960 | 0.484 | 0.849 | 0.870 | 0.491 | 0.756 |

Table 24: Test $R^2$ scores of SSA on SVHN-MNIST with different batch sizes. Higher is better

| Batch size | 8 | 16 | 32 | 64 | 128 | 256 |
|---|---|---|---|---|---|---|
| $R^2$ | $0.353_{\pm0.04}$ | $0.497_{\pm0.01}$ | $0.505_{\pm0.02}$ | $0.509_{\pm0.03}$ | $0.528_{\pm0.02}$ | $0.522_{\pm0.01}$ |

Table 25: Test $R^2$ scores of SSA on UTKFace with different batch sizes. Higher is better.

| Batch size | Defocus blur | Motion blur | Zoom blur | Contrast | Elastic transform | Jpeg compression | Pixelate | Gaussian noise | Impulse noise | Shot noise | Brightness | Fog | Snow | Mean |
|---|---|---|---|---|---|---|---|---|---|---|---|---|---|---|
| 8 | 0.779 | 0.804 | 0.796 | 0.651 | 0.867 | 0.804 | 0.914 | 0.499 | 0.516 | 0.416 | 0.834 | 0.408 | 0.390 | 0.668 |
| 16 | 0.782 | 0.809 | 0.808 | 0.699 | 0.874 | 0.807 | 0.915 | 0.545 | 0.562 | 0.476 | 0.848 | 0.421 | 0.424 | 0.690 |
| 32 | 0.792 | 0.824 | 0.830 | 0.749 | 0.885 | 0.816 | 0.929 | 0.566 | 0.583 | 0.533 | 0.858 | 0.446 | 0.482 | 0.715 |
| 64 | 0.791 | 0.830 | 0.835 | 0.769 | 0.891 | 0.822 | 0.936 | 0.569 | 0.591 | 0.540 | 0.861 | 0.446 | 0.510 | 0.722 |
| 128 | 0.788 | 0.828 | 0.835 | 0.780 | 0.893 | 0.822 | 0.939 | 0.581 | 0.590 | 0.545 | 0.862 | 0.445 | 0.523 | 0.725 |
| 256 | 0.789 | 0.823 | 0.836 | 0.770 | 0.896 | 0.821 | 0.940 | 0.540 | 0.594 | 0.551 | 0.870 | 0.439 | 0.522 | 0.723 |

Table 26: Test $R^2$ scores of SSA on Biwi Kinect with different batch sizes. Higher is better.

| | Female → Male | | | Male → Female | | | |
|---|---|---|---|---|---|---|---|
| Batch size | Pitch | Roll | Yaw | Pitch | Roll | Yaw | Mean |
| 8 | 0.872 | 0.930 | 0.450 | 0.819 | 0.770 | 0.468 | 0.718 |
| 16 | 0.870 | 0.953 | 0.463 | 0.858 | 0.857 | 0.525 | 0.754 |
| 32 | 0.868 | 0.961 | 0.498 | 0.868 | 0.886 | 0.562 | 0.774 |
| 64 | 0.859 | 0.962 | 0.515 | 0.869 | 0.889 | 0.571 | 0.777 |
| 128 | 0.841 | 0.960 | 0.524 | 0.863 | 0.883 | 0.569 | 0.773 |
| 256 | 0.819 | 0.957 | 0.521 | 0.855 | 0.870 | 0.583 | 0.768 |

Table 27: Test MAE scores on UTKFace with image corruption (lower is better). The best scores are **bolded**.

| Method | Defocus blur | Motion blur | Zoom blur | Contrast | Elastic transform | Jpeg compression | Pixelate | Gaussian noise | Impulse noise | Shot noise | Brightness | Fog | Snow | Mean |
|---|---|---|---|---|---|---|---|---|---|---|---|---|---|---|
| Source | 10.12 | 12.64 | 7.68 | 39.99 | 7.92 | 13.91 | 9.65 | 32.00 | 32.02 | 31.92 | 8.81 | 15.52 | 19.76 | 18.61 |
| DANN | 9.44 | 8.88 | 8.73 | 19.59 | 7.52 | 8.01 | 6.09 | 41.26 | 35.56 | 38.21 | 9.11 | 16.07 | 18.11 | 17.43 |
| TTT | 6.78 | 6.55 | 6.61 | 6.51 | 5.77 | 6.59 | 5.13 | 9.56 | 9.46 | 10.00 | 6.51 | 10.96 | 9.84 | 7.71 |
| BN-Adapt | 7.23 | 6.91 | 6.69 | 7.59 | 5.97 | 6.81 | 5.40 | 10.37 | 10.32 | 10.99 | 6.44 | 11.51 | 10.55 | 8.21 |
| Prototype | 21.52 | 21.62 | 21.65 | 20.00 | 21.28 | 21.02 | 21.27 | 18.46 | 18.45 | 18.42 | 21.71 | 20.71 | 20.70 | 20.52 |
| FR | 6.12 | **5.47** | 5.15 | 6.29 | 4.47 | 5.85 | **3.15** | 9.99 | 9.87 | 10.53 | 5.03 | 11.04 | 10.36 | 7.18 |
| VM | 28.28 | 28.14 | 28.42 | 27.45 | 27.75 | 27.74 | 26.97 | 29.07 | 29.21 | 29.05 | 27.80 | 29.50 | 29.19 | 28.35 |
| RSD | 6.35 | 5.68 | 5.25 | 6.60 | 4.61 | 5.93 | 3.34 | 10.26 | 10.35 | 10.94 | 5.12 | 11.13 | 9.98 | 7.35 |
| SSA (ours) | **6.05** | 5.52 | **5.09** | **6.05** | **4.46** | **5.72** | 3.27 | **9.05** | **8.95** | **9.29** | **5.01** | **10.60** | **9.50** | **6.81** |
| Oracle | 5.21 | 4.55 | 4.53 | 4.96 | 3.96 | 4.92 | 2.64 | 8.44 | 8.28 | 8.42 | 4.45 | 10.03 | 8.07 | 6.03 |

Table 28: Test MAE scores on Biwi Kinect (lower is better). The best scores are **bolded**.

| | Female → Male | | | Male → Female | | | |
|---|---|---|---|---|---|---|---|
| Method | Pitch | Roll | Yaw | Pitch | Roll | Yaw | Mean |
| Source | 0.150 | 0.081 | 0.087 | 0.160 | 0.171 | 0.093 | 0.124 |
| DANN | $0.163_{\pm0.01}$ | $0.163_{\pm0.01}$ | $0.136_{\pm0.01}$ | $0.181_{\pm0.01}$ | $0.166_{\pm0.01}$ | $0.147_{\pm0.00}$ | $0.159_{\pm0.00}$ |
| TTT | $0.283_{\pm0.02}$ | $0.236_{\pm0.00}$ | $0.140_{\pm0.00}$ | $0.165_{\pm0.00}$ | $0.212_{\pm0.00}$ | $0.190_{\pm0.00}$ | $0.205_{\pm0.00}$ |
| BN-adapt | $0.146_{\pm0.00}$ | $0.085_{\pm0.00}$ | $0.090_{\pm0.00}$ | $0.153_{\pm0.00}$ | $0.134_{\pm0.00}$ | $\mathbf{0.090}_{\pm\mathbf{0.00}}$ | $0.116_{\pm0.00}$ |
| Prototype | $6.935_{\pm0.00}$ | - | - | - | - | - | $6.935_{\pm0.00}$ |
| FR | $0.376_{\pm0.05}$ | $0.192_{\pm0.01}$ | $0.248_{\pm0.02}$ | $0.212_{\pm0.01}$ | $0.150_{\pm0.01}$ | $0.182_{\pm0.02}$ | $0.226_{\pm0.01}$ |
| VM | $0.367_{\pm0.00}$ | $0.407_{\pm0.00}$ | $0.136_{\pm0.00}$ | $0.418_{\pm0.00}$ | $0.463_{\pm0.00}$ | $0.135_{\pm0.00}$ | $0.321_{\pm0.00}$ |
| RSD | $0.142_{\pm0.01}$ | $0.085_{\pm0.00}$ | $0.090_{\pm0.00}$ | $0.154_{\pm0.00}$ | $0.132_{\pm0.00}$ | - | $0.121_{\pm0.00}$ |
| SSA (ours) | $\mathbf{0.112}_{\pm\mathbf{0.00}}$ | $\mathbf{0.079}_{\pm\mathbf{0.00}}$ | $\mathbf{0.086}_{\pm\mathbf{0.00}}$ | $\mathbf{0.141}_{\pm\mathbf{0.00}}$ | $\mathbf{0.126}_{\pm\mathbf{0.00}}$ | $\mathbf{0.090}_{\pm\mathbf{0.00}}$ | $\mathbf{0.106}_{\pm\mathbf{0.00}}$ |
| Oracle | $0.054_{\pm0.00}$ | $0.054_{\pm0.00}$ | $0.059_{\pm0.00}$ | $0.076_{\pm0.00}$ | $0.068_{\pm0.00}$ | $0.065_{\pm0.00}$ | $0.063_{\pm0.00}$ |

Table 29: Test scores on SVHN-MNIST in the batched online setting. The best scores are **bolded**.

| Method | $R^2(\uparrow)$ | RMSE ($\downarrow$) | MAE ($\downarrow$) |
|---|---|---|---|
| Source | 0.406 | 2.232 | 1.608 |
| TTT | $0.296_{\pm0.01}$ | $2.430_{\pm0.01}$ | $1.587_{\pm0.01}$ |
| BN-adapt | $0.384_{\pm0.00}$ | $2.272_{\pm0.00}$ | $1.480_{\pm0.00}$ |
| Prototype | $0.484_{\pm0.00}$ | $2.080_{\pm0.00}$ | $1.489_{\pm0.00}$ |
| FR | $0.342_{\pm0.00}$ | $2.348_{\pm0.00}$ | $1.657_{\pm0.01}$ |
| VM | $-227.105_{\pm7.22}$ | $43.729_{\pm0.69}$ | $37.021_{\pm0.72}$ |
| RSD | $0.312_{\pm0.08}$ | $2.397_{\pm0.15}$ | $1.607_{\pm0.15}$ |
| SSA (ours) | $\mathbf{0.488}_{\pm\mathbf{0.01}}$ | $\mathbf{2.072}_{\pm\mathbf{0.03}}$ | $\mathbf{1.265}_{\pm\mathbf{0.02}}$ |
| Oracle | $0.745_{\pm0.00}$ | $1.463_{\pm0.01}$ | $0.882_{\pm0.01}$ |

Table 30: Test $R^2$ scores on UTKFace with image corruption in the batched online setting. The best scores are **bolded**.

| Method | Defocus blur | Motion blur | Zoom blur | Contrast | Elastic transform | Jpeg compression | Pixelate | Gaussian noise | Impulse noise | Shot noise | Brightness | Fog | Snow | Mean |
|---|---|---|---|---|---|---|---|---|---|---|---|---|---|---|
| Source | 0.410 | 0.187 | 0.658 | −3.906 | 0.701 | 0.069 | 0.595 | −2.536 | −2.539 | −2.522 | 0.661 | −0.018 | −0.543 | −0.676 |
| TTT | 0.742 | 0.758 | 0.773 | 0.778 | 0.828 | 0.769 | 0.860 | 0.519 | 0.531 | 0.483 | 0.776 | 0.391 | 0.462 | 0.667 |
| BN-Adapt | 0.726 | 0.758 | 0.757 | 0.719 | 0.822 | 0.774 | 0.849 | 0.492 | 0.504 | 0.451 | 0.788 | 0.375 | 0.437 | 0.650 |
| Prototype | −1.001 | −1.017 | −1.014 | −0.719 | −0.965 | −0.907 | −0.972 | −0.514 | −0.513 | −0.511 | −1.003 | −0.823 | −0.822 | −0.829 |
| FR | 0.786 | **0.834** | **0.844** | 0.765 | 0.895 | 0.820 | **0.944** | 0.499 | 0.509 | 0.449 | 0.858 | 0.409 | 0.453 | 0.697 |
| VM | −1.457 | −1.401 | −1.503 | −1.230 | −1.327 | −1.343 | −1.251 | −1.538 | −1.538 | −1.524 | −1.330 | −1.632 | −1.577 | −1.435 |
| RSD | 0.783 | 0.829 | 0.843 | 0.761 | 0.893 | 0.820 | 0.940 | 0.503 | 0.509 | 0.447 | 0.858 | 0.416 | 0.483 | 0.699 |
| SSA (ours) | **0.794** | **0.834** | **0.844** | **0.789** | **0.896** | **0.823** | 0.943 | **0.550** | **0.564** | **0.524** | **0.861** | **0.427** | **0.520** | **0.721** |
| Oracle | 0.825 | 0.868 | 0.863 | 0.824 | 0.908 | 0.845 | 0.951 | 0.578 | 0.586 | 0.577 | 0.872 | 0.472 | 0.605 | 0.752 |

Table 31: Test $R^2$ scores on Biwi Kinect in the batched online setting. The best scores are **bolded**.

| Method | Female → Male | | | Male → Female | | | Mean |
|---|---|---|---|---|---|---|---|
| | Pitch | Roll | Yaw | Pitch | Roll | Yaw | |
| Source | 0.759 | 0.956 | 0.481 | 0.763 | 0.791 | 0.485 | 0.706 |
| TTT | $-0.207_{\pm 0.05}$ | $0.610_{\pm 0.00}$ | $0.010_{\pm 0.02}$ | $0.743_{\pm 0.00}$ | $0.722_{\pm 0.00}$ | $-0.296_{\pm 0.00}$ | $0.264_{\pm 0.01}$ |
| BN-adapt | $0.759_{\pm 0.00}$ | $0.951_{\pm 0.00}$ | $0.487_{\pm 0.00}$ | $0.827_{\pm 0.00}$ | $0.837_{\pm 0.00}$ | $\mathbf{0.567}_{\pm 0.00}$ | $0.738_{\pm 0.00}$ |
| Prototype | $-317.740_{\pm 0.02}$ | - | - | - | - | - | - |
| FR | $-0.124_{\pm 0.06}$ | $0.818_{\pm 0.02}$ | $-2.049_{\pm 0.24}$ | $0.775_{\pm 0.02}$ | $0.852_{\pm 0.01}$ | $0.128_{\pm 0.08}$ | $0.067_{\pm 0.06}$ |
| VM | $-0.242_{\pm 0.01}$ | $-0.057_{\pm 0.00}$ | $-0.089_{\pm 0.00}$ | $-0.101_{\pm 0.00}$ | $-0.051_{\pm 0.00}$ | $-0.006_{\pm 0.00}$ | $-0.091_{\pm 0.00}$ |
| RSD | $0.768_{\pm 0.01}$ | $0.951_{\pm 0.00}$ | $0.486_{\pm 0.00}$ | $0.826_{\pm 0.00}$ | $0.838_{\pm 0.00}$ | - | - |
| SSA (ours) | $\mathbf{0.825}_{\pm 0.00}$ | $\mathbf{0.957}_{\pm 0.00}$ | $\mathbf{0.502}_{\pm 0.00}$ | $\mathbf{0.853}_{\pm 0.00}$ | $\mathbf{0.865}_{\pm 0.00}$ | $\mathbf{0.567}_{\pm 0.00}$ | $\mathbf{0.761}_{\pm 0.00}$ |
| Oracle | $0.923_{\pm 0.00}$ | $0.971_{\pm 0.00}$ | $0.672_{\pm 0.00}$ | $0.925_{\pm 0.00}$ | $0.934_{\pm 0.00}$ | $0.717_{\pm 0.01}$ | $0.857_{\pm 0.00}$ |

