# OpenReview forum: "Test-time Adaptation for Regression by Subspace Alignment"
_ICLR.cc/2025/Conference — ICLR 2025 Poster_

### Official Review · Reviewer_gBNN · 2024-10-30

**Soundness:** 2
**Presentation:** 3
**Contribution:** 2
**Rating:** 6
**Confidence:** 4

**Summary:**

This paper investigates test-time adaptation (TTA) for regression, where a regression model pre-trained in a source domain is adapted to an unknown target distribution with unlabeled target data. To enable TTA for regression, the authors adopt a feature alignment approach, which aligns the feature distributions between the source and target domains to mitigate the domain gap. A propose Significant-subspace Alignment (SSA) is proposed for feature alignment in TTA for regression, which consists of two components: subspace detection and dimension weighting. Some experiments have been conducted to verify the performance of the proposed method.

**Strengths:**

1.	A Significant-subspace Alignment (SSA) method is proposed to address TTA for regression on the basis of the feature alignment approach.
2.	SSA consists of two components: subspace detection and dimension weighting.
3.	Subspace detection uses principal component analysis (PCA) to find a subspace and dimension weighting raises the importance of the subspace dimensions.

**Weaknesses:**

1.	The description of Figure 1 is too simple to understand the procedure of the proposed method. Some key variables (i.e., gφ and hψ) should be introduced. For easy understand, I suggest the authors provide a step-by-step description of the workflow shown in the figure.
2.	The used compared methods are all published before 2023, how the proposed method compares to or differs from some newer approaches, such as “Backpropagation-free Network for 3D Test-time Adaptation, CVPR 2024”, “Improved Self-Training for Test-Time Adaptation, CVPR, 2024”.
3.	Why no results for RSD are given in Table 3? Please provide the missing results for RSD in Table 3 or explain why these results were not included.
4.	What is the visualization effect of the proposed method? Provide T-SNE rendering.
5.	What is the limitation of the proposed method? Are there any situations where SSA might not perform well, or any assumptions it makes that might not hold in all regression tasks?

**Questions:**

Please see the weaknesses.

---

> ### Author Response · Authors · 2024-11-21
>
> Dear reviewer `gBNN`,
>
> Thank you for your careful reading.
>
>
> ## Weakness 1
>
> > The description of Figure 1 is too simple to understand the procedure of the proposed method. Some key variables (i.e., gφ and hψ) should be introduced. For easy understand, I suggest the authors provide a step-by-step description of the workflow shown in the figure.
>
>
> Thank you for your suggestion.
> We updated the figure by adding definitions of the key variables and revising the description to make our method procedure clearer in the revised paper.
>
>
>
> ## Weakness 2
>
> > The used compared methods are all published before 2023, how the proposed method compares to or differs from some newer approaches, such as “Backpropagation-free Network for 3D Test-time Adaptation, CVPR 2024”, “Improved Self-Training for Test-Time Adaptation, CVPR, 2024”.
>
> Thank you for your suggestion.
> Even in recent works, TTA for regression has not been explored yet.
> The suggested two papers are also designed for classification and include classification-specific components that cannot be rendered to regression such as entropy.
> We have compared TTA methods as much as possible in the experiment.
>
> We added the suggested papers in the related work (**Section 5.3**) in the revised paper.
>
>
>
> ## Weakness 3
>
> > Why no results for RSD are given in Table 3? Please provide the missing results for RSD in Table 3 or explain why these results were not included.
>
> Thank you for your clarification.
> Although we tried RSD on the California Housing dataset, it was numerically unstable and did not work.
> As we discussed in **Section 4.3.2**, this is because RSD requires SVD on the target feature batch in every iteration.
> We observed that SVD resulted in convergence error on the California Housing dataset.
>
>
> ## Weakness 4
>
> > What is the visualization effect of the proposed method? Provide T-SNE rendering.
>
> Thank you for your suggestion.
> In **Figure 4** of **Appendix D.2**, we visualized the features by PCA.
> We also computed the optimal transport distance (OTD) between the source and target features in the principal subspace and found that our SSA produces a smaller OTD, i.e., a smaller feature gap.
>
> Here, we added UMAP visualization of top $K=100$ principal components in **Figure 5** to see the relations not limited within top-2 principal components.
> We used UMAP instead of t-SNE because UMAP can first learn the mappng of the source featurs before TTA, which are shared among the methods, and then map additional features.
> We mapped the target features after TTA using the learned UMAP mapping.
> The target feature distribution becomes closer to the source one with our SSA.
>
>
> ## Weakness 5
>
> > What is the limitation of the proposed method? Are there any situations where SSA might not perform well, or any assumptions it makes that might not hold in all regression tasks?
>
> Thank you for your clarification.
> As we discussed in **Appendix A**, one limitation is that our SSA assumes a covariate shift, where $p(y|x)$ does not change.
> For distribution shifts where $p(y|x)$ changes, e.g., concept drift, we will need not only to align $p(z)$ but also to address the change of the label distribution.
> For concept drift, one may combine label shift adaptation techniques such as [1] and [2].
> Addressing concept drift is one direction of future work.
>
> [1] Park et al., Label Shift Adapter for Test-Time Adaptation under Covariate and Label Shifts, ICCV2023.
> [2] Sun et al., Beyond Invariance: Test-Time Label-Shift Adaptation for Distributions with "Spurious" Correlations, NeurIPS2023.

---

### Official Review · Reviewer_xN4r · 2024-11-01

**Soundness:** 3
**Presentation:** 3
**Contribution:** 3
**Rating:** 8
**Confidence:** 4

**Summary:**

The paper proposes a test-time domain adaptation method for regression. They show that learned embeddings are more compressed in regression than classification. To address this, they introduce Significant-subspace Alignment (SSA). SSA uses PCA to detect a significant feature subspace where features are concentrated, then aligns the first two moments of source embeddings and target batch within this subspace. They also apply dimension weighting to prioritize subspace dimensions based on their significance to the output. The proposed method is evaluated on four regression datasets and demonstrates superior performance compared to baseline TTA methods designed for classification tasks, and domain adaptation for regression adapted for TTA .

**Strengths:**

- The paper proposes a novel method for Test time domain adaptation for regression, and adapted common benchmarking procedures for classification to regression, especially on the UTK Face dataset.

- The paper is well-written and clear.

- The method and results seem convincing and easy to implement.

**Weaknesses:**

- In the related work, feature alignment methods for TTA are reviewed with the claim, “Although some of these methods are directly applicable to regression, we have observed that they are not effective or even degrade regression performance.” However, these methods are not included in the experiments, leaving this claim unsupported.
- In Table 1, the subspace dimension for Biwi Kinect is shown as ‘34.5’, which is unclear, as dimensions are typically integers.
- MLPs and CNNs seem to produce embeddings with different ranks, with MLPs showing less low-rank behavior. A discussion or remark on this difference could provide useful insights.
- The test setup lacks clarity on whether SSA is applied in an online or episodic adaptation manner, and it’s unclear if batches are fed uniformly.
- The dimension weighting approach may lead to disproportionately high contributions from small-scale features if they are assigned high weights (when learned on source), potentially resulting in values similar to large-scale features with lower weights (when learned on source).   Did you consider any normalization or balancing mechanism to prevent this scaling effect?

**Questions:**

See Weaknesses

---

> ### Author Response · Authors · 2024-11-21
>
> Dear reviewer `xN4r`,
>
> Thank you for your insightful comments.
>
>
> ## Weakness 1
>
> > In the related work, feature alignment methods for TTA are reviewed with the claim, “Although some of these methods are directly applicable to regression, we have observed that they are not effective or even degrade regression performance.” However, these methods are not included in the experiments, leaving this claim unsupported.
>
> Thank you for pointing it out.
> We have already included feature alignment methods that can be directly rendered to regression as much as possible: BN-adapt and FR as shown in **Tables 2-5**.
> We have observed they degrade the performance when rendered to regression.
> The other feature alignment methods include classification-specific components that cannot be applied to regression, e.g., sample weighting by confidence in DELTA[1], class-wise feature statistics in CFA[2] and CAFA[3], and class-categorical prediction in CAFe[4].
>
>
> [1] Zhao et al., DELTA: degradation-free fully test-time adaptation, ICLR2023.
> [2] Kojima et al., Robustifying vision transformer without retraining from scratch by test-time class-conditional feature alignment, IJCAI2022.
> [3] Jung et al., CAFA: Class-Aware Feature Alignment for Test-Time Adaptation, ICCV2023.
> [4] Adachi et al., Covariance-aware feature alignment with pre-computed source statistics for test-time adaptation to multiple image corruptions, ICIP2023.
>
>
> ## Weakness 2
>
> > In Table 1, the subspace dimension for Biwi Kinect is shown as ‘34.5’, which is unclear, as dimensions are typically integers.
>
> Thank you for your clarification.
> For Biwi Kinect, we computed the numbers of the feature dimensions for the six models trained on the six combinations of the domain and targets, i.e., {male, female} $\times$ {pitch, roll, yaw}.
> We reported the mean of the six numbers of feature dimensions.
> We clarified this point in the revised paper.
> The actual number of the feature dimensions of each model is as follows:
>
> **Table A**: The number of feature dimensions on Biwi Kinect.
> | Source domain | Target | \#Valid dims. | \#Subspace dims. |
> |--|--|--|--|
> | Male | Pitch | 677 | 33 |
> | Male | Yaw | 735 | 12 |
> | Male | Roll | 640 | 39 |
> | Female | Pitch | 699 | 40 |
> | Female | Yaw | 823 | 34 |
> | Female | Roll | 704 | 49 |
>
>
> ## Weakness 3
>
> > MLPs and CNNs seem to produce embeddings with different ranks, with MLPs showing less low-rank behavior. A discussion or remark on this difference could provide useful insights.
>
> Thank you for pointing it out.
> For the California Housing dataset, we used an MLP with only five layers, which has a lower capacity than the ResNets used for the other datasets.
> One explanation for the MLP's less low-rank behavior is that the complexity of the task is high relative to the model capacity.
> For instance, when training the ResNet-26, which has high capacity, on the SVHN, which has low complexity, the subspace dimension is much smaller than the original dimensions.
>
> We added the explanation to **Section 4.3.1** in the revised paper.

---

> > ### Author Response · Authors · 2024-11-21
> >
> > ## Weakness 4
> >
> > > The test setup lacks clarity on whether SSA is applied in an online or episodic adaptation manner, and it’s unclear if batches are fed uniformly.
> >
> > Thank you for your clarification.
> > For fair comparison with other methods, we evaluated in an offline manner, i.e., we ran TTA for one epoch and then ran evaluation, and batches are uniformly sampled.
> > This manner is widely adopted in existing TTA works [5,6,7].
> >
> > But our method can also run in an batched online manner, i.e., model update and evaluation are alternatively performed in every iteration.
> > Here, we show the summary of batched online result, which our method also consistently ourperformed the baselines:
> >
> > **Table B**: Test $R^2$ scores on SVHN-MNIST in an batched online manner.
> > | Method | $R^2$ |
> > |--|--|
> > | Source | ${0.406}$ |
> > | TTT | ${0.296}$ |
> > | BN-adapt | ${0.384}$ |
> > | Prototype | ${0.484}$ |
> > | FR | ${0.342}$ |
> > | VM | ${-227.105}$ |
> > | RSD | ${0.312}$ |
> > | SSA (ours) | $\mathbf{0.488}$ |
> > | Oracle | ${0.745}$ |
> >
> > **Table C**: Test $R^2$ scores on UTKFace in an batched online manner (averaged over corruption types).
> > | Method | $R^2$ |
> > |--|--|
> > | Source | $-0.678$ |
> > | TTT | $0.667$ |
> > | BN-adapt | $0.650$ |
> > | Prototype | $-0.829$ |
> > | FR | $0.697$ |
> > | VM | $-1.435$ |
> > | RSD | $0.699$ |
> > | SSA (ours) | $\mathbf{0.721}$ |
> > | Oracle | $0.752$ |
> >
> > **Table D**: Test $R^2$ scores on Biwi Kinect in an batched online manner (averaged over gender/task combinations).
> > | Method | $R^2$ |
> > |--|--|
> > | Source | $0.706$ |
> > | TTT | $0.264$ |
> > | BN-adapt | $0.738$ |
> > | Prototype | - |
> > | FR | $0.067$ |
> > | VM | $-0.091$ |
> > | RSD | - |
> > | SSA (ours) | $\mathbf{0.761}$ |
> >
> > We revised our paper by clarifying the experimental settings in **Appendix C.3** and adding the results for batched online setting in **Tables 29 to 31** of **Appendix D.10.**
> >
> >
> > [5] Wang et al., Tent: Fully Test-time Adaptation by Entropy Minimization, ICLR2021.
> > [6] Zhou et al., Bayesian Adaptation for Covariate Shift, NeurIPS2021.
> > [7] Eastwood et al., Source-Free Adaptation to Measurement Shift via Bottom-Up Feature Restoration, ICLR2022.
> >
> >
> > ## Weakness 5
> >
> > > The dimension weighting approach may lead to disproportionately high contributions from small-scale features if they are assigned high weights (when learned on source), potentially resulting in values similar to large-scale features with lower weights (when learned on source). Did you consider any normalization or balancing mechanism to prevent this scaling effect?
> >
> > Thank you for pointing it out.
> > As shown in **Table 7**, we checked the correlation between the feature scale along the subspace dimension $\lambda^\text{s}_d$ and weight $\alpha_d$, and found that they are correlated with, but not completely proportional.
> > This means there are dimensions with small feature scales but highly important (and vice versa).
> > Thus, assigning high weights to small-scale features is as we intended.
> >
> > For normalization, one may use whitening instead of PCA to balance the scaling effect.
> > Developing a more effective way of dimension weighting is one direction of our future work.

---

> > > ### Comment · Reviewer_xN4r · 2024-11-24
> > >
> > > Dear Authors,
> > >
> > > I acknowledge your response and am happy with the current version of the manuscript.

---

> > > > ### Author Response · Authors · 2024-11-28
> > > >
> > > > Thank you for raising the score!
> > > > We again appreciate your comments and insights, which definitely improved our paper.

---

### Official Review · Reviewer_66U8 · 2024-11-03

**Soundness:** 3
**Presentation:** 2
**Contribution:** 3
**Rating:** 6
**Confidence:** 2

**Summary:**

This paper investigates the application of Test-Time Adaptation for regression tasks, where a regression model pre-trained in a source domain is adapted to an unknown target distribution using unlabeled target data. The authors note that most existing TTA methods are designed for classification tasks and do not directly apply to regression models, which typically output single scalar values rather than class-categorical predictions. To address this, the paper proposes a feature alignment approach called Significant-subspace Alignment, which consists of two components: subspace detection and dimension weighting. SSA aims to align feature distributions between the source and target domains to mitigate the domain gap, focusing on a representative and significant subspace of the feature space. Experimental results on various real-world datasets demonstrate that SSA outperforms several baseline methods.

**Strengths:**

1.SSA introduces a novel approach for TTA in regression tasks by combining subspace detection and dimension weighting, which is an innovative contribution to the field.
2.The paper conducts extensive experiments on multiple real-world datasets, validating the effectiveness of SSA.
3.Compared to the original model and other baseline methods, SSA achieves higher R2 scores across multiple datasets, demonstrating performance improvements in regression tasks.

**Weaknesses:**

1.SSA assumes covariate shift, where p(y|x) remains unchanged. The paper does not address distribution shifts where p(y|x) changes, such as concept drift, limiting its applicability in broader scenarios.
2.While the paper mentions the selection of the parameter λ, it lacks a detailed discussion on the impact of other hyperparameters, which could affect the model's generalizability and adaptability.
3.The paper does not discuss the computational complexity of SSA, particularly its performance with large-scale datasets, which is crucial for practical applications.

**Questions:**

How does SSA perform in multi-task and multi-class regression tasks? Can it be extended to these scenarios?

---

> ### Author Response · Authors · 2024-11-21
>
> Dear reviewer `66U8`,
>
> Thank you for your thoughtful feedback.
>
> ## Weakness 1
>
> > SSA assumes covariate shift, where p(y|x) remains unchanged. The paper does not address distribution shifts where p(y|x) changes, such as concept drift, limiting its applicability in broader scenarios.
>
>
> Thank you for pointing it out.
> As we discussed in **Appendix A**, one limitation is that our SSA assumes a covariate shift, where $p(y|x)$ does not change.
> But covariate shift is also one realistic distribution shift that has been actively studied in TTA [1].
> For distribution shifts where $p(y|x)$ changes, e.g., concept drift, we will need not only to align $p(z)$ but also to address the change of the label distribution.
> For concept drift, one may combine label shift adaptation techniques such as [2] and [3].
> Addressing concept drift is one direction of future work.
>
> [1] Liang et al., A Comprehensive Survey on Test-Time Adaptation under Distribution Shifts, IJCV, 2023.
> [2] Park et al., Label Shift Adapter for Test-Time Adaptation under Covariate and Label Shifts, ICCV2023.
> [3] Sun et al., Beyond Invariance: Test-Time Label-Shift Adaptation for Distributions with "Spurious" Correlations, NeurIPS2023.
>
>
> ## Weakness 2
>
> > While the paper mentions the selection of the parameter λ, it lacks a detailed discussion on the impact of other hyperparameters, which could affect the model's generalizability and adaptability.
>
> Thank you for pointing it out.
> Although the only hyperparameter of our method is the number of subspace dimensions $K$, we performed additional investigation on the learning rate and batch size, which are important hyperparameters in practice.
> The following tables show the results:
>
> **Table A**: Impact of the learning rate on SVHN-MNIST
> | Learning rate | $R^2$ |
> |--|--|
> | 0.0001 | ${0.472}\_{\pm 0.00}$ |
> | 0.0005 | ${0.516}\_{\pm 0.01}$ |
> | 0.001 | ${0.509}\_{\pm 0.03}$ |
> | 0.005 | ${0.510}\_{\pm 0.07}$ |
> | 0.01 | ${0.212}\_{\pm 0.21}$ |
>
> **Table B**: Impact of the batch size on SVHN-MNIST
> | Batch size | $R^2$ |
> |--|--|
> | 8 | ${0.353}\_{\pm 0.04}$ |
> | 16 | ${0.497}\_{\pm 0.01}$ |
> | 32 | ${0.505}\_{\pm 0.02}$ |
> | 64 | ${0.509}\_{\pm 0.03}$ |
> | 128 | ${0.528}\_{\pm 0.02}$ |
> | 256 | ${0.522}\_{\pm 0.01}$ |
>
> **Table C**: Impact of the learning rate on UTKFace (averaged over corruption types)
> | Learning rate | $R^2$ |
> |--|--|
> | 0.0001 | ${0.703}\_{\pm 0.00}$ |
> | 0.0005 | ${0.715}\_{\pm 0.00}$ |
> | 0.001 | ${0.722}\_{\pm 0.00}$ |
> | 0.005 | ${0.706}\_{\pm 0.00}$ |
> | 0.01 | ${0.658}\_{\pm 0.00}$ |
>
> **Table D**: Impact of the batch size on UTKFace (averaged over corruption types)
> | Batch size | $R^2$ |
> |--|--|
> | 8 | ${0.668}\_{\pm 0.00}$ |
> | 16 | ${0.690}\_{\pm 0.01}$ |
> | 32 | ${0.715}\_{\pm 0.00}$ |
> | 64 | ${0.722}\_{\pm 0.00}$ |
> | 128 | ${0.725}\_{\pm 0.00}$ |
> | 256 | ${0.723}\_{\pm 0.00}$ |
>
> **Table E**: Impact of the learning rate on Biwi Kinect (averaged over genders and targets)
> | Learning rate | $R^2$ |
> |--|--|
> | 0.0001 | ${0.753}\_{\pm 0.00}$ |
> | 0.0005 | ${0.772}\_{\pm 0.00}$ |
> | 0.001 | ${0.777}\_{\pm 0.00}$ |
> | 0.005 | ${0.772}\_{\pm 0.00}$ |
> | 0.01 | ${0.756}\_{\pm 0.00}$ |
>
> **Table F**: Impact of the batch size on Biwi Kinect (averaged over genders and targets)
> | Batch size | $R^2$ |
> |--|--|
> | 8 | ${0.718}\_{\pm 0.02}$ |
> | 16 | ${0.754}\_{\pm 0.00}$ |
> | 32 | ${0.774}\_{\pm 0.00}$ |
> | 64 | ${0.777}\_{\pm 0.00}$ |
> | 128 | ${0.773}\_{\pm 0.00}$ |
> | 256 | ${0.768}\_{\pm 0.00}$ |
>
> As we can observe from the tables, typical ranges of the learning rate and batch size produce competitive performance.
> For the batch size, a larger batch size results in better performance since the estimation of feature mean and variance becomes more accurate.
> But batch sizes $\geq 16$ produce competitive performance.
>
> We also added more detailed results, i.e., corruption-wise on UTKFace and gender/target-wise on Biwi Kinect, in **Tables 21-26** of **Appendix D.8** in the revised paper.
>
>
> ## Weakness 3
>
> > The paper does not discuss the computational complexity of SSA, particularly its performance with large-scale datasets, which is crucial for practical applications.
>
> Thank you for pointing it out.
> Although benchmarking on large-scale datasets is crutial for practical applications, there are no large-scale regression datasets including distribution shifts like ImageNet family in classification.
> As our method can run in an batched online setting as shown in **Tables 29-31** in **Appendix D.10** of the revised paper, we believe that our method can scale up to large-scale datasets.
> Also, one may combine continual learning techniques like EWC [4] to stabilize adaptation.
> Constructing large-scale benchmarks for regression TTA and scaling up our method is future work.
>
> [4] Kirkpatrick et al., Overcoming catastrophic forgetting in neural networks.

---

> > ### Author Response · Authors · 2024-11-21
> >
> > ## Question
> >
> > > How does SSA perform in multi-task and multi-class regression tasks? Can it be extended to these scenarios?
> >
> > Thank you for your clarification.
> > We can extend our SSA to multi-target scenarios since the feature alignment can be used regardless of output forms.
> >
> > We have tried multi-task regression with Biwi Kinect in **Table 17** of **Appendix D.6**, demonstrating that our method improved the $R^2$ score.
> > We have also tried classification in **Tables 19 and 20** of **Appendix D.7**.
> > The results show that our SSA can improve not only regression but also classification.
> > Moreover, combining SSA with existing classification TTA further improves the accuracy.

---

> > > ### Comment · Reviewer_66U8 · 2024-11-25
> > >
> > > Dear Authors，
> > > The paper's contributions to the field of test-time adaptation for regression tasks have been further reinforced by the additional experiments and analyses, and your well-written clarifications and explanations of the assumptions and limitations of the SSA method have been appreciated.

---

> > > > ### Author Response · Authors · 2024-12-04
> > > >
> > > > Thank you for your careful reading and thoughtful comments.
> > > > We appreciate that the contributions of our work are recognized.

---

### Official Review · Reviewer_t9WM · 2024-11-04

**Soundness:** 3
**Presentation:** 2
**Contribution:** 2
**Rating:** 6
**Confidence:** 4

**Summary:**

This paper addresses a crucial yet under-explored area in machine learning: test-time adaptation (TTA) for regression models. The authors rightly highlight the limitations of existing TTA methods, which are primarily tailored for classification tasks, and propose a novel approach termed Significant-subspace Alignment (SSA). The paper presents a clear rationale for the need to adapt TTA techniques for regression and offers a compelling solution through feature alignment.

**Strengths:**

The topic of adapting regression models to unknown target distributions is highly relevant, particularly given the increasing use of machine learning in diverse applications. The proposed SSA approach is innovative and addresses a gap in the literature.

**Weaknesses:**

1. In Section 3.1 of the article, the use of "two diagonal Gaussian distributions" is not adequately justified, and the advantages and limitations of employing such distributions are not discussed.
2. The proposed method in this paper exhibits limited originality, as its various components are commonly found in existing models.
3. The dataset utilized in this study has a relatively small sample size, which weakens the persuasiveness of the findings. Additionally, there is a lack of comparative algorithms from the past three years, resulting in insufficient theoretical validation. Furthermore, the measurement standard is solely based on R²; incorporating additional metrics such as RMSE and RMAE would enhance the evaluation.

**Questions:**

1. Why are "two diagonal Gaussian distributions" used in Chapter 3? What is the rationale behind this choice?
2. In Section 4.1, the first dataset utilizes a classification dataset as a regression task. Is this approach reasonable and justifiable? Regarding the second dataset, is it appropriate to consider a noisy version of the test set as the target domain?
3. Could you provide a proof or evidence demonstrating the effectiveness of Significant-subspace Alignment (SSA)?

---

> ### Author Response · Authors · 2024-11-21
>
> Dear reviewer `t9WM`,
>
> Thank you for your constructive comments.
>
> ## Weakness 1 and Question 1
>
> > In Section 3.1 of the article, the use of "two diagonal Gaussian distributions" is not adequately justified, and the advantages and limitations of employing such distributions are not discussed.
>
> > Why are "two diagonal Gaussian distributions" used in Chapter 3? What is the rationale behind this choice?
>
>
> Thank you for your clarification.
> Assuming Gaussian distributions is reasonable for the following reasons:
>
> First, we can easily compute the KL divergence in a closed form by assuming the diagonal Gaussian distribution, as in Equation (3).
> Second, as shown in **Figure 3**, features follow a Gaussian-like distribution when projected onto the feature subspace detected by subspace detection.
> This is due to the central limit theorem described in **Section 4.3.4**, i.e., the features are more likely to follow a Gaussian distribution in the subspace as the number of the original feature dimensions increases.
> Moreover, since subspace detection uses the PCA, the features projected onto the subspace are decorrelated.
> Thus, assuming that each dimension is independent, i.e., diagonal, is also reasonable.
>
> We added explanation to use diagonal Gaussian to **Section 3.2** of the revised paper.
>
> One limitation of employing Gaussian distributions is that features are not likely to follow Gaussian distributions when the number of dimensions of the original feature space is small.
> In such a case, one may employ another distribution, e.g., a histogram used in BUFR [1].
> Addressing such a case is one direction of future work.
>
> [1] Eastwood et al., Source-Free Adaptation to Measurement Shift via Bottom-Up Feature Restoration. ICLR2022.
>
>
> ## Weakness 2
>
> > The proposed method in this paper exhibits limited originality, as its various components are commonly found in existing models.
>
> Thank you for pointing it out.
> Our contributions include the finding that features of a trained regression model are distributed in only a small subspace.
> This finding is novel, and we believe that our method is also simple yet novel since it is based on the finding.
> Moreover, incorporating the idea of feature alignment in the subspace for TTA is also novel.
>
>
> ## Weakness 3
>
> > The dataset utilized in this study has a relatively small sample size, which weakens the persuasiveness of the findings.
>
> Thank you for pointing it out.
> We have evaluated methods on various regression datasets with distribution shifts as much as possible.
> Since TTA for regression is an unexplored field, constructing common regression benchmarks including large-scale datasets is an important future work.
>
> > there is a lack of comparative algorithms from the past three years, resulting in insufficient theoretical validation.
>
> This is because there are no comparable baselines in our regression TTA setting except for BN-adapt and FR.
> We have compared methods that can be directly rendered to regression as much as possible.
> This is the first study that explicitly addresses TTA for regression, and our method can serve as a good baseline for future work in this setting.
>
> Although we have discussed TTA methods from the past three years in **Section 5.3**, they include classification-specific components that cannot be rendered to regression, e.g., sample weighting by confidence in DELTA[2], class-wise feature statistics in CFA[3] and CAFA[4], and class-categorical prediction in CAFe[5].
>
>
> [2] Zhao et al., DELTA: degradation-free fully test-time adaptation, ICLR2023.
> [3] Kojima et al., Robustifying vision transformer without retraining from scratch by test-time class-conditional feature alignment, IJCAI2022.
> [4] Jung et al., CAFA: Class-Aware Feature Alignment for Test-Time Adaptation, ICCV2023.
> [5] Adachi et al., Covariance-aware feature alignment with pre-computed source statistics for test-time adaptation to multiple image corruptions, ICIP2023.
>
> > the measurement standard is solely based on R²; incorporating additional metrics such as RMSE and RMAE would enhance the evaluation.
>
> Thank you for your suggestion.
> We added MAE to **Tables 2, 3, 27 and 28** for SVHN-MNIST, California Housing, UTKFace, and Biwi Kinect, respectively, in the revised paper.
> Our method consistently outperformed the baselines also in the additional metric.

---

> > ### Author Response · Authors · 2024-11-21
> >
> > ## Question 2
> >
> > > In Section 4.1, the first dataset utilizes a classification dataset as a regression task. Is this approach reasonable and justifiable? Regarding the second dataset, is it appropriate to consider a noisy version of the test set as the target domain?
> >
> > Thank you for your clarification.
> > We think using SVHN and MNIST for regression is reasonable since one can easily reproduce this task like MNIST in classification.
> > We also used real-world datasets in the experiment, which demonstrates higher performance than the baselines.
> >
> >
> > For UTKFace, we employed image corruption for evaluating on various types of distribution shifts since the other datasets mainly includes domain shifts.
> > Existing studies on TTA [2,3,4] conventionally evaluate methods on image corruption such as ImageNet-C and CIFAR10/100-C.
> > We decided to use UTKFace with adding corruption because there is no regression dataset including image corruption.
> >
> > [2] Wang et al., Tent: Fully Test-time Adaptation by Entropy Minimization, ICLR2021.
> > [3] Niu et al., Efficient Test-Time Model Adaptation without Forgetting, ICML2022.
> > [4] Zhao et al., DELTA: degradation-free fully test-time adaptation, ICLR2023.
> >
> >
> > ## Question 3
> >
> > > Could you provide a proof or evidence demonstrating the effectiveness of Significant-subspace Alignment (SSA)?
> >
> > Thank you for your clarification.
> > For a theoretical perspective, as we discussed in **Section 3.1**, our method is based on the fact that a KL divergence term between the source and target feature distributions is included in the upper bound of the target loss [5].
> > For the empirical effectiveness of SSA, we have empirically observed a significant improvement from the ablation study in **Table 6**.
> >
> > [5] Nguyen et al., KL Guided Domain Adaptation, ICLR2022.

---

> > ### Comment · Reviewer_t9WM · 2024-12-02
> >
> > Thanks for response, I have updated my score.

---

> > > ### Author Response · Authors · 2024-12-02
> > >
> > > Thank you for raising the score! We again appreciate your insightful comments, which definitely improved our paper.

---

### Author Response · Authors · 2024-11-21

We thank all the reviewers for their constructive feedbacks and suggestions, which have helped us improve the quality and clarity of our work.

Based on the reviewers' comments, we revised our paper as follows:

- Added discussion on employing diagonal Gaussian distribution in **Section 3.2** based on reviewer `t9WM`'s comment.
- Added evaluation in MAE in **Tables 2 and 3** of **Section 4.3.2** and **Tables 27 and 28** of **Appendix D.9.** based on reviewer `t9WM`'s comment.
- Added experiments with different learning rate and batch size in **Tables 21-26** in **Appendix D.8** based on reviewer `66U8`'s comment.
- Clarified the experimental setting in comparing the number of the feature dimensions in **Table 1** based on reviewer `xN4r`'s comment.
- Added explanation on the difference of the subspace dimensions between CNN and MLP in **Section 4.3.1** based on reviewer `xN4r`'s comment.
- Clarified the experimental setting in **Appendix C.3** and added results of batched online setting in **Tables 29-31** in **Appendix D.10** based on reviewer `xN4r`'s comment.
- Updated the overview of our SSA in **Figure 1** based on reviewer `gBNN`'s comment.
- Added citations of related papers in **Section 5.3** based on reviewer `gBNN`'s comment.
- Added feature visualization by UMAP in **Figure 5** based on reviewer `gBNN`'s comment.

We hope that the revised paper helps the reviewers to re-evaluate our work.

---

### Public Comment · ~Kazuki_Adachi1 · 2025-03-19

The code is available at https://github.com/kzkadc/regression-tta

---

### Meta-Review · Area_Chair_bAf6 · 2024-12-20

**Metareview:**

After extensive author-reviewer discussions, all reviewers suggest that the paper should be accepted. In this case, I am happy to recommend accepting the paper.

**Additional Comments On Reviewer Discussion:**

NA

---

### Decision · Program_Chairs · 2025-01-22

Accept (Poster)